# JPEG INSPIRED DEEP LEARNING

**Ahmed H. Salamah**\*, **Kaixiang Zheng**\*, **Yiwen Liu & En-Hui Yang**
Department of Electrical and Computer Engineering, University of Waterloo
{ahamsalamah,k56zheng,e3liu,ehyang}@uwaterloo.ca

## ABSTRACT

Although it is traditionally believed that lossy image compression, such as JPEG compression, has a negative impact on the performance of deep neural networks (DNNs), it is shown by recent works that well-crafted JPEG compression can actually improve the performance of deep learning (DL). Inspired by this, we propose JPEG-DL, a novel DL framework that prepends any underlying DNN architecture with a trainable JPEG compression layer. To make the quantization operation in JPEG compression trainable, a new differentiable soft quantizer is employed at the JPEG layer, and then the quantization operation and underlying DNN are jointly trained. Extensive experiments show that in comparison with the standard DL, JPEG-DL delivers significant accuracy improvements across various datasets and model architectures while enhancing robustness against adversarial attacks. Particularly, on some fine-grained image classification datasets, JPEG-DL can increase prediction accuracy by as much as 20.9%. Our code is available on https://github.com/AhmedHussKhalifa/JPEG-Inspired-DL.git.

## 1 INTRODUCTION

JPEG compression (Pennebaker & Mitchell, 1992) is the *defacto* lossy image compression technique with ubiquitous presence in real-world applications. With the development of deep learning, more and more images, potentially compressed by JPEG, are consumed by deep neural networks (DNNs). Naturally, it's of interest to study how JPEG compression will impact DNN performance for computer vision tasks, and extensive research has been conducted along this line such as Dodge & Karam (2016); Liu et al. (2018) and Xie & Kim (2019). These initial explorations establish a widely accepted view that the information loss caused by JPEG obscures important features in the input image, thereby negatively impacting DNN performance.

However, it was shown by Yang et al. (2021) that the above conventional wisdom does not hold anymore if JPEG compression is applied intelligently and adaptively on a per-image basis. Indeed, Yang et al. (2021) showed that if applied appropriately, JPEG compression can actually improve DNN performance, at least in theory. They further proposed to train a DNN based on images with various JPEG quality levels. While they managed to demonstrate improved performance with a specially designed DNN topology, the model is too cumbersome to be fully trained, leading to suboptimal performance. Moreover, the adherence to the default JPEG quantization also limits the effectiveness of this method. On the other hand, given a fixed DNN model, Zheng et al. (2023) and Salamah et al. (2024b) found that its performance could be slightly improved if the input images got compressed by JPEG with optimized quantization parameters. However, the performance gain is not significant due to the frozen DNN model. Although these existing works provide promising insights to improve DNN performance with JPEG compression, a solid solution that can fully unleash the potential of this idea has yet to be found.

---

\*Authors contributed equally.

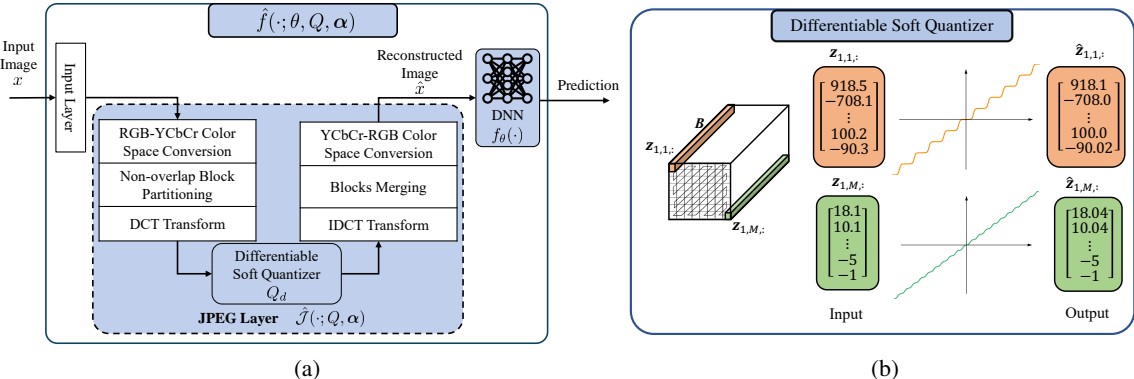

Figure 1: (a) The JPEG-DL framework consists of a JPEG layer followed by a standard DNN, where the standard forward/inverse processes of JPEG depicted in white boxes are fixed and not trained. The JPEG pipeline receives a conventionally preprocessed input image $x \in \mathbb{R}^{3 \times W \times H}$ sampled from the underlying task's dataset. The JPEG layer, equipped with a differentiable soft quantizer, and the underlying DNN form a unified new DNN architecture, which are shown in blue indicating that they are trainable components. (b) As an example, we show $z_{1,:,:}$, i.e., the DCT representation of the Y channel of an image, by a tensor consisting of $B$ blocks of DCT coefficients. Each $8 \times 8$ block contains $M = 64$ DCT frequencies, ordered from low to high in a zigzag manner. Then, we show how $z_{1,1,:}$ and $z_{1,M,:}$ are quantized by $Q_d(\cdot\ ; q = 1, \alpha = 10)$ and $Q_d(\cdot\ ; q = 0.5, \alpha = 16)$, respectively.

To address the above issue, in this paper, we propose jointly optimizing both JPEG quantization operation and a DNN to achieve greater effectiveness. To this end, we first introduce a trainable JPEG compression layer, the structure of which is illustrated in Fig. 1a. This layer follows the standard JPEG pipeline to convert a preprocessed input image, after passing through the input layer, into blockwise Discrete Cosine Transform (DCT) coefficients in the YCbCr color space. A novel differentiable soft quantizer ($Q_d$) is then applied to quantize the DCT coefficients at each frequency position, followed by the standard JPEG inverse process to reconstruct the RGB image. We then present JPEG-DL, a novel deep learning (DL) framework that introduces a new DNN architecture by inserting a JPEG layer directly after the original input layer of any underlying DNN architecture. This layer can be considered an integral part of any underlying DNN architecture, forming a new unified DNN architecture, whose parameters get optimized jointly with DNN model weights during training. The core of JPEG-DL is $Q_d$ as it substitutes the non-differentiable, hard quantization in JPEG with a differentiable, soft quantization operation defined by a neat analytical formula, which not only facilitates gradient-based optimization for quantization parameters but also introduces additional trainable non-linearity to the overall image understanding pipeline. To validate the effectiveness of JPEG-DL, we conducted extensive experiments for image classification on six datasets including four fine-grained classification datasets (Wah et al., 2011; Khosla et al., 2011; Nilsback & Zisserman, 2008; Parkhi et al., 2012), CIFAR-100 (Krizhevsky et al., 2009) and ImageNet (Deng et al., 2009). Results show that JPEG-DL significantly and consistently outperforms the standard DL across various DNN architectures, with a negligible increase in model complexity. Specifically, JPEG-DL improves classification accuracy by up to 20.9% on some fine-grained classification dataset, while adding only 128 trainable parameters to the DL pipeline. Moreover, the superiority of JPEG-DL over the standard DL is further demonstrated by the enhanced adversarial robustness of the learned unified architecture.

The main contributions of this paper can be summarized as follows:

- We introduce a novel trainable JPEG layer leveraging differentiable soft quantizers with nice analytical formulas.
- Based on the new JPEG layer, we propose a new DL framework dubbed JPEG-DL, which jointly optimizes the JPEG layer and the DNN model during training.
- The outstanding performance of JPEG-DL over the standard DL is verified by comprehensive experimental results on various image classification datasets across multiple DNN architectures and for a variety of tasks including adversarial defense.

## 2    BACKGROUND AND RELATED WORK

**JPEG applications.**  JPEG was originally developed as a lossy image compression technique based on transform coding, which reduces image file sizes by generating their compact representations. Beyond its traditional use, JPEG has found numerous applications in deep learning: (1) it has been utilized as a data augmentation technique to improve robustness of DNNs against image compression (Benbarrad et al., 2022); (2) it has been employed as an empirical defense method against adversarial attacks, effectively reducing adversarial perturbations and enhancing the adversarial robustness of DNNs (Dziugaite et al., 2016; Das et al., 2017; Guo et al., 2018); and (3) it has been integrated into the knowledge distillation (Hinton et al., 2015) framework by Salamah et al. (2024a), where it helps the teacher model transfer knowledge to the student model in a more effective way. In contrast, this paper focuses on leveraging JPEG to enhance the natural performance, instead of the robust performance, of DNNs without relying on any teacher model.

**Optimizing JPEG Compression for DNN vision.**  As a lossy image compression technique, JPEG is developed specifically for the human visual system. As a result, while the information loss introduced by JPEG is often imperceptible to humans, it can significantly degrade the performance of DNNs. This issue gives rise to a line of research which optimizes JPEG compression based on DNN perception. For instance, given a pretrained DNN, Xie & Kim (2019), Zheng et al. (2023) and Salamah et al. (2024b) first derive its sensitivity to different DCT frequencies, based on which they customize JPEG quantization tables for this DNN to reach the optimal rate-accuracy tradeoff. Another popular direction is to make JPEG trainable, which integrates the JPEG encoder and DNN into an end-to-end differentiable training framework (Luo et al., 2020; Xie et al., 2022). These methods create differentiable proxies for image quantization and bitrate calculation, so that JPEG compression can be optimized via backpropagation in order to minimize a total loss considering both DNN performance and bitrate. However, all the above works focus on mitigating DNN performance degradation in the presence of JPEG compressed images, but offer little to no improvement on DNN performance when raw images are given as input. On the contrary, this paper leverages JPEG purely as a tool to improve DNN performance on raw images, regardless of its compression capability.

**Improving DNN performance with JPEG.** Due to the aforementioned reason, JPEG compression generally hurts DNN performance. However, it's recently demonstrated by Yang et al. (2021) that, with an oracle guiding the compression process, one can select an optimal quality level to compress each image, enabling the DNN to make its best possible prediction for it. By applying this adaptive JPEG compression across a set of images, one can actually improve the DNN prediction accuracy considerably. This phenomenon, termed "compression helps" in the original paper, is justified by the fact that compression can remove noise and disturbing background features, thereby highlighting the main object in an image, which helps DNNs make better prediction. Due to the need of ground truth labels in the compression stage, the above adaptive compression scheme is not realizable in most real world applications; however, the discovery of "compression helps" at least show the potential of using JPEG to improve DNN performance. Thus motivated, the authors in turn proposed an implementable way to improve DNN performance with JPEG. They built a new DNN topology that incorporates 11 parallel branches of an underlying pretrained model, with each branch obtain-

ing as input either the raw image or its compressed version at a varying quality level. The penultimate layer representations from all branches are concatenated and fed into a classification head, which is then trained together with those 11 model backbones as a unified DNN structure. This new DNN topology is clearly too complex to train, so the authors opted for partial training to mitigate the training complexity, therefore resulting in a suboptimal performance. In contrast, our proposed method introduces only 128 additional trainable parameters to the existing DL pipeline, causing negligible complexity increase. Moreover, it's noteworthy that our method is orthogonal to theirs, as our trainable JPEG quantization tables can be embedded into their framework to replace the default JPEG quantization tables, thereby further improving the performance.

**Trainable Activation Functions.** Relentless efforts have been made to search for activation functions that can improve DNN performance. Among all the directions, trainable activation functions have gained particular interest for their flexibility, expressive power, and adaptability during training. Chen & Chang (1996) propose the adjustable generalized hyperbolic tangent function, which extends the classic hyperbolic tangent function by introducing parameters to control the saturation level and slope of the function. He et al. (2015) introduce the parametric ReLU (PReLU), a variant of ReLU with a trainable parameter that adjusts the negative part of ReLU. More recently, Kolmogorov-Arnold Networks (KANs) (Liu et al., 2024) employ trainable activation functions on edges, with nodes simply summing all the incoming activations. Interestingly, the differentiable soft quantizers in our JPEG layer can be interpreted as trainable activation functions whose input is the DCT representations of images. These trainable soft quantizers effectively introduce additional nonlinearity to DNN models, thus improving their expressive power.

## 3 JPEG-DL: JPEG INSPIRED DL

### 3.1 PROBLEM FORMULATION

In the JPEG pipeline, an RGB image $x \in \mathbb{R}^{3 \times W \times H}$ is first converted to the YCbCr color space and then partitioned into $B$ non-overlapping $8 \times 8$ blocks, where DCT is applied to each of these blocks to obtain the corresponding DCT coefficients. For each color channel, the DCT coefficients in each block are then flattened following the zigzag order, resulting in $M = 64$ frequency positions ordered from low frequency to high frequency. Therefore, we denote the DCT coefficients of the image $x$ as $\boldsymbol{z} = [z_{l,m,n}]$, where $l = 1, 2, 3$ corresponds to the color channel Y, Cb and Cr respectively, $1 \leq m \leq M$ is the index for frequency position, and $1 \leq n \leq B$ is the index for block. Up to this point, all operations involved are differentiable. Next, quantization tables $Q_Y = [q_1, q_2, \ldots, q_M]$ and $Q_C = [q_{M+1}, q_{M+2}, \ldots, q_{2M}]$ are used for the luminance (Y) and chrominance (CbCr) channels respectively to quantize their DCT coefficients. Following uniform quantization, we obtain quantized DCT coefficients $\hat{z}_{l,m,n} = \lfloor z_{l,m,n}/q_m \rceil \cdot q_m$ for $l = 1$, and $\hat{z}_{l,m,n} = \lfloor z_{l,m,n}/q_{M+m} \rceil \cdot q_{M+m}$ for $l = 2, 3$. Note that quantization is non-differentiable due to the use of the rounding operation. Finally, inverse operations including blockwise inverse DCT (IDCT) transform, blocks merging and YCbCr-to-RGB color space conversion are conducted over $\hat{z}$ sequentially to obtain the reconstructed RGB image $\hat{x}$. Similar to their forward counterparts, all these inverse operations are differentiable. Denoting the composition of all the above operations as $\mathcal{J}$, we then have $\hat{x} = \mathcal{J}(x; Q)$, where $Q = (Q_Y, Q_C)$. Hereafter, the mapping $\mathcal{J}$ stands for the JPEG encoding-decoding operation.

In supervised learning, each $x \in \mathcal{X}$ corresponds to a ground truth label $y \in \mathcal{Y}$. Let $f_\theta$ represent a DNN model with trainable weights $\theta$, and let $\mathcal{L}$ denote the loss function used to train this DNN. In standard DL, the primary objective is to solve the following minimization problem:

$$\min_\theta \ \mathbb{E}[\mathcal{L}(f_\theta(x), y)]. \tag{1}$$

In contrast, JPEG-DL tries to improve the performance of DNN by jointly training it with the JPEG operation. As a result, the formulation should be instead:

$$\min_{\theta, Q} \ \mathbb{E}[\mathcal{L}(f_\theta(\mathcal{J}(x; Q)), y)]. \tag{2}$$

However, in order to solve (2) with gradient descent, the key challenge is caused by the non-differentiable quantization operation, which makes the gradients w.r.t. $Q$ almost zero everywhere. To address this issue, we will introduce a *differentiable soft quantizer* ($Q_d$) in the next subsection, replacing the uniform quantizer ($Q_u$) used in $\mathcal{J}$.

## 3.2 DIFFERENTIABLE SOFT QUANTIZER

Denote the index set of uniform quantization as

$$\mathcal{A} = \{-L, -L+1, \ldots, 0, \ldots, L-1, L\}. \tag{3}$$

For convenience, $\mathcal{A}$ is also regarded as a vector of length $2L+1$. Multiplying $\mathcal{A}$ with a quantization step size $q$, we get the corresponding reconstruction space

$$\hat{\mathcal{A}} = q \times [-L, -L+1, \ldots, 0, \ldots, L-1, L]. \tag{4}$$

Again, we will regard $\hat{\mathcal{A}}$ as both a vector and a set.

To randomly quantize a DCT coefficient $z$ to an element in $\hat{\mathcal{A}}$, we invoke from Yang & Hamidi (2024) a trainable conditional probability mass function (CPMF) $P_\alpha(\cdot|z)$ over the reconstruction space $\hat{\mathcal{A}}$ or equivalently the index set $\mathcal{A}$ given $z$, where $\alpha > 0$ is a trainable parameter:

$$P_\alpha(iq|z) = \frac{e^{-\alpha(z-iq)^2}}{\sum_{j\in\mathcal{A}} e^{-\alpha(z-jq)^2}}, \ \forall i \in \mathcal{A}. \tag{5}$$

Extend $z$ to a vector of length $2L+1$, i.e., $[z]_{2L+1} = [\ \overbrace{z, \ldots, z}^{2L+1 \text{ times}}\ ]$. Then, the CPMF $P_\alpha(\cdot|z)$, regarded as a vector of length $2L+1$, can be easily computed via the softmax operation $\sigma(\cdot)$:

$$\left[P_\alpha(\cdot|z)\right]_{2L+1} = \sigma\Big(-\alpha \times \big([z]_{2L+1} - \hat{\mathcal{A}}\big)^2\Big). \tag{6}$$

With the CPMF $P_\alpha(\cdot|z)$, $z$ is now quantized to each $iq \in \hat{\mathcal{A}}$ with probability $P_\alpha(iq|z)$. Note that as $\alpha \to \infty$, $P_\alpha(\cdot|z)$ approaches an one-hot vector with probability 1 at the nearest point to $z$ in $\hat{\mathcal{A}}$ and 0 elsewhere. Therefore, the resulting random quantizer effectively functions as the deterministic uniform quantizer $Q_u(z) = \lfloor z/q \rfloor \cdot q$.

Based on the CPMF $P_\alpha(\cdot|z)$, we can now define a differentiable soft quantizer $Q_d$ as the conditional expectation of $iq$ given $z$, i.e.,

$$Q_d(z) = \mathbb{E}[iq|z] = \sum_{i\in\mathcal{A}} P_\alpha(iq|z) \cdot iq. \tag{7}$$

Similarly, as $\alpha \to \infty$, $Q_d$ also goes to $Q_u$. Fig. 2 shows how the shape of $Q_d$ varies w.r.t $\alpha$, given a fixed $q$.

This soft quantizer $Q_d$ serves as an analytical proxy for $Q_u$. It's differentiable everywhere, allowing gradients to flow through it smoothly. More importantly, compared to $Q_u$, $Q_d$ involves a trainable parameter $\alpha$ which can adjust the softness of the quantizer, thereby introducing more flexibility. In view of these nice properties, $Q_d$ is the ideal candidate in place of $Q_u$ used in $\mathcal{J}$.

As a side note, for the reader who is not familiar with quantization, but familiar with the attention operation used in transformer models (Vaswani, 2017), $Q_d$ can be regarded as an attention operation in broad sense, with the query being $z$, the key and value being $\hat{\mathcal{A}}$, and the similarity metric between the query and key being negative squared distance instead of dot product.

### 3.3 OVERALL FRAMEWORK OF JPEG-DL

Substituting $Q_u$ in $\mathcal{J}$ with $Q_d$, we get a differentiable JPEG layer $\hat{\mathcal{J}}$ parameterized by $Q$ and $\boldsymbol{\alpha}$, where $\boldsymbol{\alpha} = (\boldsymbol{\alpha}_Y, \boldsymbol{\alpha}_C)$. $\boldsymbol{\alpha}_Y = [\alpha_1, \alpha_2, \ldots, \alpha_M]$ and $\boldsymbol{\alpha}_C = [\alpha_{M+1}, \alpha_{M+2}, \ldots, \alpha_{2M}]$ are $\alpha$ tables for the luminance and chrominance channels respectively, used in conjunction with $Q_Y$ and $Q_C$ to quantize DCT coefficients. Following the proposed soft quantization, we obtain quantized DCT coefficients $\hat{z}_{l,m,n} = Q_d(z_{l,m,n}; q_m, \alpha_m)$ for $l = 1$, and $\hat{z}_{l,m,n} = Q_d(z_{l,m,n}; q_{M+m}, \alpha_{M+m})$ for $l = 2, 3$, where $Q_d(z; q, \alpha)$ denotes a differentiable soft quantizer defined in (7) parameterized by a quantization step $q$ and a scaling factor $\alpha$. Overall, for an input image $x$, we have $\hat{x} = \hat{\mathcal{J}}(x; Q, \boldsymbol{\alpha})$. Therefore, we can rewrite (2), the JPEG-DL formulation, as

$$\min_{\theta, Q, \boldsymbol{\alpha}} \ \mathbb{E}[\mathcal{L}(f_\theta(\hat{\mathcal{J}}(x; Q, \boldsymbol{\alpha})), y)], \tag{8}$$

where the expectation can be approximated by the empirical mean over a mini-batch in actual training. Thanks to the use of $Q_d$, (8) can now be solved by gradient descent with ease (see Appendix A.1 for the derivative analytical formulas).

After training with JPEG-DL, we get the optimized parameters $\theta^*$, $Q^*$ and $\boldsymbol{\alpha}^*$. Then, we consider the composition of the JPEG layer and the underlying DNN as a unified DNN model $\hat{f}(x; \theta^*, Q^*, \boldsymbol{\alpha}^*) = f_{\theta^*}(\hat{\mathcal{J}}(x; Q^*, \boldsymbol{\alpha}^*))$ to do validation. Concretely, any raw input $x$ should be fed into $\hat{\mathcal{J}}$, instead of directly to $f_{\theta^*}$, thus allowing the reconstructed image $\hat{x}$ to be fed into the underlying DNN $f_{\theta^*}$. In other words, the JPEG layer is prepended as the first layer of any underlying DNN.

To conclude this section, we refer readers to Fig. 1a and Fig. 1b for an illustration of the inner workings of the JPEG-DL framework.

## 4 EXPERIMENTS

**CIFAR-100.** We evaluate our proposed method using a transformer-based architecture and four state-of-the-art convolutional neural networks (CNNs): EfficientFormer-L1 (Li et al., 2022), ResNet (He et al., 2016), VGG (Simonyan & Zisserman, 2014), MobileNet (Sandler et al., 2018), and ShuffleNet (Ma et al., 2018). For ResNet, we employ CIFAR-specific versions: ResNet32, ResNet56, and ResNet110. For VGG, we utilize VGG8 and VGG13. All CNN architectures follow the training recipe from CRD (Tian et al., 2020) (see Appendix A.3), while for the transformer-based architecture, EfficientFormer-L1, we adhere to the setup proposed by Xu et al. (2023) (see Appendix A.5).

**Fine-grained Tasks.** These tasks involve visually similar classes and typically feature fewer training samples per class compared to conventional classification tasks. We evaluate our method on four datasets: CUB-200-2011 (Wah et al., 2011), Stanford Dogs (Khosla et al., 2011), Flowers (Nilsback & Zisserman, 2008), and Pets (Parkhi et al., 2012). For CNN architectures, we employ PreAct ResNet-18 (He et al., 2016)

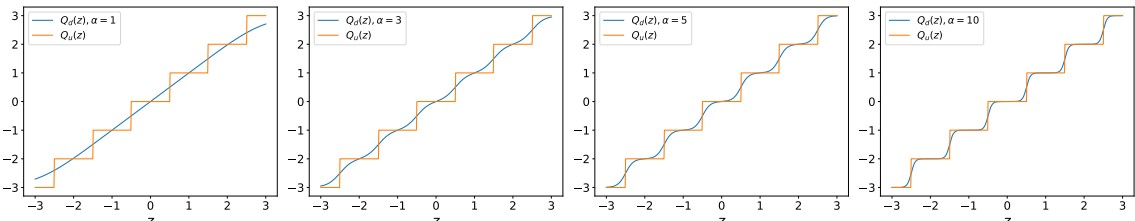

Figure 2: Illustration of $Q_u$ vs. $Q_d$ with $\alpha = 1, 3, 5, 10$, where $L$ and $q$ are set to 3 and 1, respectively.

and DenseNet-BC (Huang et al., 2017), following the experimental setup and architecture modifications of Zhang et al. (2017). For the transformer-based architecture, we use EfficientFormer-L1, adopting the setup outlined by Xu et al. (2023) (see Appendices A.4 and A.5).

**JPEG-layer settings** for CIFAR-100 and fine-grained tasks. To train our JPEG layer, we first study the gradient nature of $Q_d$ with respect to $q$ and $\alpha$ in (5) by changing one parameter and fixing the other. Both $q$ and $\alpha$ are sharable parameters among different numbers of blocks per image. During the calculation of the gradient of $Q_d$ with respect to $\alpha$, our analysis shows that when $\alpha$ is sufficiently large, such that $Q_d$ is not too far from $Q_u$, the gradient magnitude for $\alpha$ is almost zero, as shown in Appendix A.1. This indicates that $\alpha$ will not be updated effectively if initialized within a reasonable range. Experiments also confirmed that making $\alpha$ trainable does not significantly impact model performance. As a result, we choose not to train $\alpha$ in our framework. However, when we explore the calculation of the gradient of $Q_d$ with respect to $q$, the accumulated gradient received from different blocks per image during backpropagation results in unstable gradient magnitudes at a fixed value of $\alpha$, as shown in our analysis in Appendix A.1. To address this, we use the ADAM optimizer, which adapts the learning rate for each trainable parameter $q$ in our JPEG layer, enabling more efficient training and better convergence. For CIFAR-100, we set the learning rate to 0.003 across all tested models. For the fine-grained datasets, we set the JPEG learning rate to 0.005. Across these datasets, we fix $\alpha_m = 5$ for all $1 \leq m \leq 2M$, and set $L = 2^{b-1}$ in (3), where $b$ is a tunable hyperparameter set to 8.

**ImageNet-1K.** For all experiments on this dataset, we utilize the standard training recipes shown by Paszke et al. (2019) without any modifications. We use SqueezeNet (Iandola, 2016), ResNet-18 and ResNet-34 as our testing underlying models.

**JPEG-layer settings** for ImageNet-1K. We will utilize specific settings to control the gradient magnitude to ensure more stable updates for $Q$. We define the Gradient Scaling Constants $\hbar_m = \alpha_m q_m^2$, $1 \leq m \leq 2M$, which allows us to control the magnitude of gradients w.r.t $Q$. Specifically, we fix $\hbar_m = 0.7$ for all $1 \leq m \leq 2M$. During training, we update the value of $\alpha_m$ to be $\hbar_m/q_m^2$ before calculating the gradients w.r.t $Q$. As a result of controlling the maximum gradient magnitude, we can optimize $Q$ using an SGD optimizer with 0.5 learning rate, instead of an ADAM optimizer. This approach ensures stable and efficient training for quantization table updates (see Appendix A.2 for more details). For this dataset, $b$ is equal to 11 across all tested models.

**Quantization Table Initialization.** For CIFAR-100 and fine-grained tasks, we initialize the quantization table $Q$ based on the reciprocal of sensitivity for each DCT frequency, given a pre-trained model of the underlying DNN architecture, following the approach described by Zheng et al. (2023); Salamah et al. (2024b). Broadly speaking, the sensitivity of a frequency indicates the rate of change of the loss function w.r.t. the perturbation on this frequency, so we favor a smaller quantization step for a more sensitive frequency to limit the distortion amount on it. For ImageNet-1K, we adopt the strategy from Esser et al. (2019), where the initialization of $q_m$ is based on the average of absolute values of DCT coefficients across all blocks in all training images that will be quantized by $q_m$. Specifically, $q_m = 2 \sum_{k=1}^N \sum_{n=1}^B |z_{1,m,n}^{(k)}|/(NB\sqrt{2^{b-1}})$ for $1 \leq m \leq M$, i.e. $Q_Y$, and $q_m = \sum_{k=1}^N \sum_{l=2}^3 \sum_{n=1}^B |z_{l,m,n}^{(k)}|/(NB\sqrt{2^{b-1}})$ for $M+1 \leq m \leq 2M$, i.e., $Q_C$, where $k$ is the index for image and $N$ is the number of training images.

Table 1: Top-1 validation accuracy (%) for Baseline and JPEG-DL on CIFAR-100. The Baseline results are from Tian et al. (2020). For JPEG-DL, we report the mean and standard deviation of experimental results over three runs.

| Method | Res32 | Res56 | Res110 | VGG8 | VGG13 | MobileNetV2 | ShuffleNetV2 |
|---|---|---|---|---|---|---|---|
| Baseline | 71.14 | 72.34 | 73.79 | 70.36 | 73.77 | 64.6 | 71.82 |
| JPEG-DL | **71.92**$_{\pm 0.31}$ (+0.78) | **73.39**$_{\pm 0.19}$ (+1.05) | **74.46**$_{\pm 0.11}$ (+0.67) | **71.10**$_{\pm 0.41}$ (+0.74) | **75.32**$_{\pm 0.10}$ (+1.55) | **65.91**$_{\pm 0.11}$ (+1.31) | **73.04**$_{\pm 0.16}$ (+1.22) |

Table 2: Top-1 validation accuracy (%) on various fine-grained image classification tasks and model architectures. We report the mean and standard deviation of experimental results over three runs.

| Model | Method | CUB-200 | Dogs | Flowers | Pets |
|---|---|---|---|---|---|
| ResNet-18 | Baseline | $54.00_{\pm1.43}$ | $63.71_{\pm0.32}$ | $57.13_{\pm1.28}$ | $70.37_{\pm0.84}$ |
| | JPEG-DL | $\mathbf{58.81}_{\pm0.12}$ (+4.81) | $\mathbf{65.57}_{\pm0.37}$ (+1.86) | $\mathbf{68.76}_{\pm0.57}$ (+11.63) | $\mathbf{74.84}_{\pm0.66}$ (+4.47) |
| DenseNet-121 | Baseline | $57.70_{\pm0.44}$ | $66.61_{\pm0.17}$ | $51.32_{\pm0.57}$ | $70.26_{\pm0.79}$ |
| | JPEG-DL | $\mathbf{61.32}_{\pm0.43}$ (+3.62) | $\mathbf{69.67}_{\pm0.58}$ (+3.06) | $\mathbf{72.22}_{\pm1.05}$ (+20.90) | $\mathbf{75.90}_{\pm0.68}$ (+5.64) |

Table 3: Top-1 validation accuracy (%) on ImageNet with different model architectures.

| Method | SqueezeNetV1.1 | Resnet18 | Resnet34 |
|---|---|---|---|
| Baseline | 57.95 | 69.75 | 73.31 |
| JPEG-DL | $\mathbf{58.26}$ (+0.31) | $\mathbf{70.13}$ (+0.38) | $\mathbf{73.54}$ (+0.23) |

**CIFAR-100 and Fine-grained tasks Results.** The performance of JPEG-DL is shown in Tables 1 and 2. Across all seven tested models for CIFAR-100, JPEG-DL consistently provides improvements, with gains of up to 1.53% in top-1 accuracy. In the fine-grained tasks, JPEG-DL offers a substantial performance increase, with improvements of up to 20.90% across all datasets using two different models. Additional results for CIFAR-100 and fine-grained tasks using a transformer-based model are shown in Appendices A.5. We further extend our results on fine-grained tasks to include a comparison with additional baselines that address the non-differentiability and zero derivative problems of JPEG quantization (see Appendix A.6). After demonstrating the nonlinearity introduced by inserting the JPEG layer after the input layer, we extend our evaluation to include the replacement of the ReLU activation function with our JPEG layer, which highlights the potential of new forms of nonlinear operations in DNN architectures (see Appendix A.7). Furthermore, we provide comprehensive experiments on the impact of chrominance subsampling on JPEG-DL, as well as a comparison with both non-learnable and learnable preprocessing methods (see Appendix A.8).

**ImageNet-1K Results.** The performance of JPEG-DL is shown in Table 3. With a trivial increase in complexity (adding 128 parameters), JPEG-DL achieves a gain of 0.31% in top-1 accuracy for SqueezeNetV1.1 compared to the baseline using a single round of $Q_d$ quantization operation. By increasing the number of quantization rounds to five, we observe an additional improvement of 0.20%, leading to a total gain of 0.51% over the baseline. The best results are indicated in bold, and values in parentheses indicate relative accuracy gains over the baseline.

## 5 ANALYSIS AND DISCUSSION

**Robustness.** JPEG has generally been used as an empirical defense mechanism against adversarial attacks by mitigating adversarial perturbations and enhancing the robustness of DNNs, as discussed in Section 2. To evaluate the adversarial robustness of JPEG-DL models in comparison to standard DNN, we conduct experiments using two attack methods, FGSM and PGD, on CIFAR-100 with two different models from Table 1. The perturbation budget, Epsilon, ranged from 1 to 4 for both attack methods. For PGD, we applied 5 steps with a perturbation step size of $(2.5 \times \text{Epsilon})/\text{steps}$, following the setup used by Madry (2017). As shown in Fig. 3, the JPEG-DL models significantly improve the adversarial robustness compared to the standard DNN models, with improvements of up to 15% for FGSM and 6% for PGD. We will show some examples of the quantization tables used in this study in the next subsection.

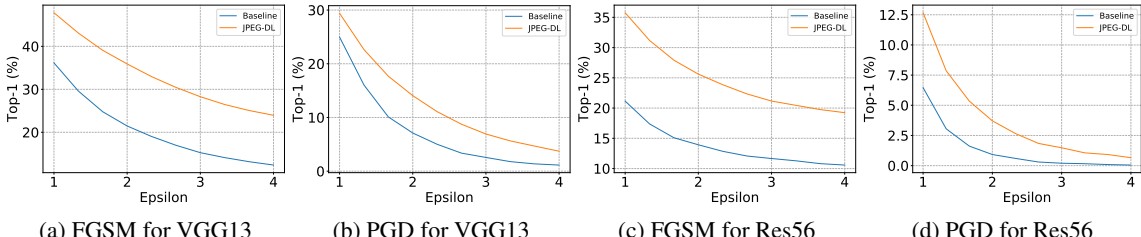

(a) FGSM for VGG13      (b) PGD for VGG13      (c) FGSM for Res56      (d) PGD for Res56

Figure 3: Evaluate the adversarial robustness of JPEG-DL models in comparison to standard DNN on VGG13 and Res56 for CIFAR-100 against FGSM and PGD attacks.

**Designed Quantization Tables.** Fig. 4 presents the Y and CbCr quantization tables, both at initialization and after convergence, for VGG13 trained on CIFAR-100 and ResNet-18 trained on the CUB200 dataset. The estimated sensitivity values used to initialize these quantization tables are provided in Appendix A.10.

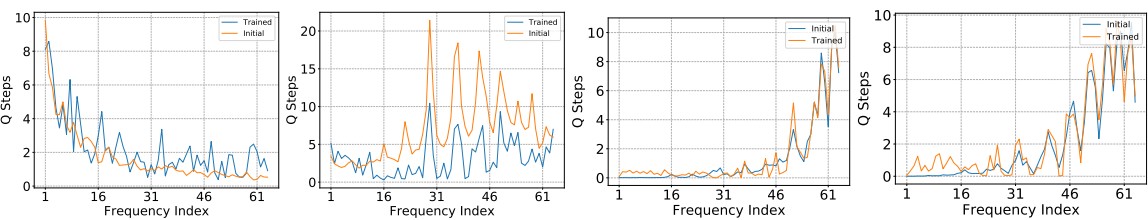

(a) VGG13 (Y Channel)    (b) VGG13 (CbCr Channel)    (c) ResNet-18(Y Channel)    (d) ResNet-18 (CbCr Channel)

Figure 4: Initial and final quantization tables for VGG13 trained on CIFAR-100 and ResNet18 trained on CUB200, with frequency indices arranged in the default zigzag order.

**Feature maps visualization.** Fig. 5 presents the feature maps extracted after the first dense block in DenseNet-121 for both the JPEG-DL model and the baseline model, trained on the CUB200 dataset using the model from Table 2. The output of the feature maps at this stage is of size 56×56, and both sets are shown in the same sequence using the same original image. The shown example was incorrectly classified by the baseline model, while the JPEG-DL model correctly classified it. In this figure, it is evident that the feature maps from the JPEG-DL model show significantly better contrast between the foreground information (the bird) and the background compared to the feature maps generated by the baseline model. Specifically, the foreground object in the JPEG-DL feature maps is enclosed within a well-defined contour, making it visually distinguishable from the background. In contrast, the baseline model's feature maps show a more blended structure, where the foreground contains higher energy in low frequencies, causing it to blend more smoothly with the background. Additionally, another example in Appendix A.11 shows a similar phenomenon in addition to background information being more effectively removed, following the same setup as the first example. These discrepancies in feature map clarity and contrast are propagated through subsequent blocks of DenseNet-121, eventually contributing to the misclassification problem observed in the baseline model.

**Interpretability using CAM.** Fig. 6 illustrates GradCAM++ visualizations (Chattopadhay et al., 2018) for two examples from the CUB-200 dataset, comparing the baseline model with JPEG-DL using ResNet-18 as our underlying model. In both instances, the baseline model incorrectly classifies the input, while the JPEG-DL model correctly classifies it. The visualizations highlight how JPEG-DL focuses more precisely

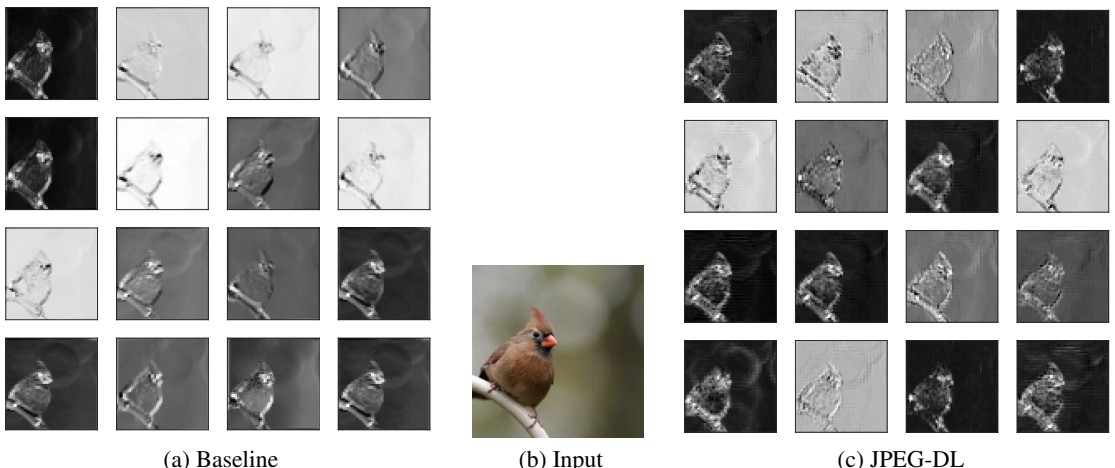

Figure 5: Feature maps of size 56×56 are shown after the first dense block in DenseNet-121 for both JPEG-DL and baseline models Figs. in 5a and 5c, respectively, using an original input shown in Fig.5b. The JPEG-DL model highlights the foreground (bird) more distinctly, while the baseline model shows less contrast, contributing to its misclassification.

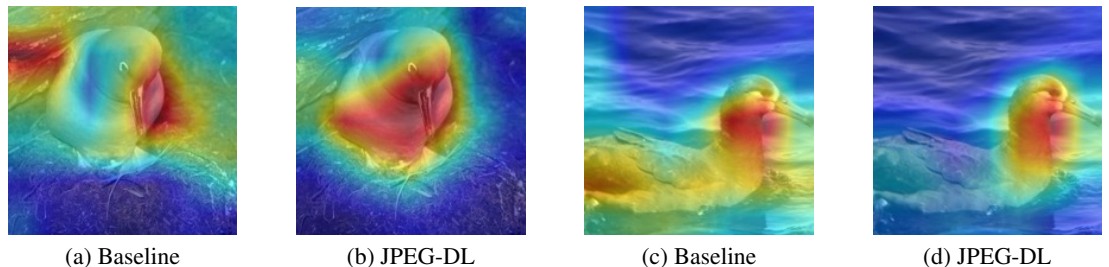

Figure 6: GradCAM++ visualization for baseline and JPEG-DL models on CUB200 using two examples, where the baseline model incorrectly classified them and the JPEG-DL model correctly classified them.

on the main object in the image, demonstrating the model's improved attention to key regions that contribute to correct classification. This highlights the effectiveness of JPEG-DL in enhancing model interpretability and performance, and it also supports the case for background removal discussed in the previous subsection.

## 6 CONCLUSION

In contrast to the conventional understanding that JPEG compression negatively impacts the DL performance, this paper introduces a novel trainable JPEG compression layer into the DL pipeline to improve DL performance, termed JPEG-DL. This layer enables gradient-based optimization for quantization parameters and can be integrated seamlessly with any underlying DNN to be trained jointly. To validate the effectiveness of JPEG-DL, we conduct extensive experiments on six image classification datasets and show significant gains over the standard DL. Additionally, we demonstrate that JPEG-DL improves the adversarial robustness of the learned model.

ACKNOWLEDGMENTS

This work was supported in part by the Natural Sciences and Engineering Research Council of Canada under Grant RGPIN203035-22, and by the Canada Research Chairs Program.

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

# A APPENDIX

## A.1 PARTIAL DERIVATIVES OF $Q_d$

With the CPMF $P_\alpha(\cdot|z)$ shown in (6), $z$ is now quantized to each $\hat{z} \in \hat{\mathcal{A}}$ with probability $P_\alpha(\hat{z}|z)$. Denote this random mapping by

$$\hat{z} = Q_{\mathrm{p}}(z). \tag{9}$$

Note that given $z$, $\hat{z}$ is a random variable taking values in $\hat{\mathcal{A}}$ with distribution $P_\alpha(\cdot|z)$.

### A.1.1 QUANTIZATION STEP AND INPUT

The partial derivatives of $Q_d(z)$ w.r.t. $q$ and $z$ are obtained as

$$\frac{\partial Q_d(z)}{\partial z} = 2\alpha \mathrm{Var}\{Q_p(z)\}, \tag{10a}$$

$$\begin{aligned}\frac{\partial Q_d(z)}{\partial q} = \frac{1}{q}\Big(&\mathbb{E}\{Q_p(z)\} + (2\alpha z)\mathrm{Var}\{Q_p(z)\}\\ &- (2\alpha)\mathrm{Skew_u}\{Q_p(z)\}\Big),\end{aligned} \tag{10b}$$

where for any random variable $V$,

$$\mathrm{Skew_u}(V) \triangleq \sum_v v^3 \mathbb{P}_V(v) - \Big(\sum_v v\mathbb{P}_V(v)\Big)\Big(\sum_v v^2\mathbb{P}_V(v)\Big).$$

### A.1.2 SCALING FACTOR $\alpha$

The partial derivatives of $Q_d(Z)$ w.r.t. $\alpha$ obtained as

$$\begin{aligned}\frac{\partial Q_d(z)}{\partial \alpha} &= \frac{\partial \sum_{i\in\mathcal{A}} iqP_\alpha(iq|z)}{\partial \alpha}\\ &= \sum_{i\in\mathcal{A}} iq\frac{\partial P_\alpha(iq|z)}{\partial \alpha}\end{aligned} \tag{11}$$

$$\begin{aligned}\frac{\partial P_\alpha(iq|z)}{\partial \alpha} &= \frac{-(z-iq)^2 e^{-\alpha(z-iq)^2}}{\sum_{j\in\mathcal{A}} e^{-\alpha(z-jq)^2}} + \frac{e^{-\alpha(z-iq)^2}\sum_{j\in\mathcal{A}}(z-jq)^2 e^{-\alpha(z-jq)^2}}{(\sum_{j\in\mathcal{A}} e^{-\alpha(z-jq)^2})^2}\\ &= P_\alpha(iq|z)\sum_{j\in\mathcal{A}}(z-jq)^2 P_\alpha(jq|z) - (z-iq)^2 P_\alpha(iq|z)\end{aligned} \tag{12}$$

Plugging (2) in (1) yields

$$\begin{aligned}\frac{\partial Q_d(z)}{\partial \alpha} &= \sum_{i\in\mathcal{A}} iqP_\alpha(iq|z)\sum_{j\in\mathcal{A}}(z-jq)^2 P_\alpha(jq|z) - \sum_{i\in\mathcal{A}} iq(z-iq)^2 P_\alpha(iq|z)\\ &= \mathbb{E}\{Q_p(z)\}\mathbb{E}\{(z-Q_p(z))^2\} - \mathbb{E}\{Q_p(z)(z-Q_p(z))^2\}\\ &= -Cov\{Q_p(z), (z-Q_p(z))^2\}\end{aligned} \tag{13}$$

To gain a better understanding, we fix $q$ and $b$, and analyzed how $\frac{\partial Q_d(z)}{\partial \alpha}$ behaves for different values of $\alpha$. From Fig. 7, it is evident that the gradient magnitude decreases as $\alpha$ increases. This indicates that $Q_d$ approaches the shape of the uniform quantizer $Q_u$, as shown previously in Fig. 2, leading to a reduction in the gradient magnitude for $\alpha$. As a result, when $\alpha$ becomes sufficiently large, the gradient approaches zero, effectively preventing further updates to $\alpha$, making it act as a non-trainable parameter at a certain point of the training process. Based on this observation and verified experimental results, we opt not to train $\alpha$ for simple image classification tasks like CIFAR-100 and fine-grained datasets, as it does not impact the overall performance.

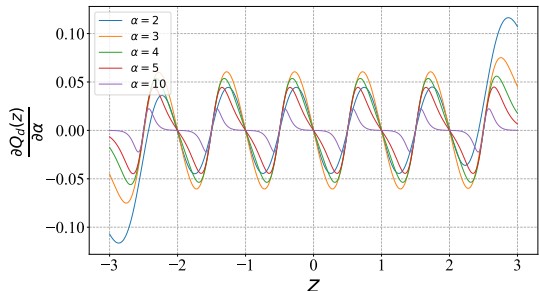

Figure 7: This Figure illustrates the partial derivatives of $Q_d(z)$ w.r.t. the scaling factor $\alpha$, where $b = 3$ and $q = 1$.

## A.2 GRADIENT SCALING

Figures 8a and 8b demonstrate the impact of varying $\alpha$ and $q$ on the gradient magnitude of $Q_d(z)$ with respect to $q$. As observed, decreasing the value of $q$ while keeping $\alpha$ constant leads to a decrease in the gradient magnitude. Similarly, decreasing the value of $\alpha$ while maintaining a fixed $q$ also results in a reduction of the gradient magnitude. To address the instability in updating the trainable parameters $Q$, we propose utilizing $\alpha$ to regulate the magnitude of $\frac{\partial Q_d(z)}{\partial q}$, as illustrated in Figure 8b. By leveraging this control mechanism of adjusting $\alpha$, we can stabilize the magnitude of the gradients that update $q$. This behavior suggests a relationship between $q$ and $\alpha$ in controlling the gradient magnitude of $Q_d(z)$.

To explore the relationship between $\alpha$ and $q$, we refer to the exponent in (5), $\alpha(z - iq)^2$, which can be rewritten by expressing $z = cq$, resulting in the form $\alpha q^2 (c - i)^2$. From this, we define a new term, $\hbar = \alpha q^2$, referred to as the *Gradient Scaling Constant*. To further illustrate this relationship, in Fig. 9, we set $\hbar = 2$, and by selecting different pairs of $\alpha$ and $q$ values, we demonstrate that the maximum magnitude of $\frac{\partial Q_d(z)}{\partial q}$ remains invariant. This confirms that the gradient magnitude can be effectively controlled by adjusting $\alpha$ based on the last updated value of $q$, according to the specified $\hbar$ value. This adjustment allows for controlled quantization updates, reducing the potential instability during training. Moreover, by controlling the gradient magnitude, we simplify the optimization process, enabling the use of a single learning rate for all $q$ values by using the SGD optimizer, instead of the ADAM optimizer. This gradient scaling mechanism is analogous to the ADAM optimizer, which adapts different learning rates for individual trainable parameters based on momentum and recent gradient magnitudes.

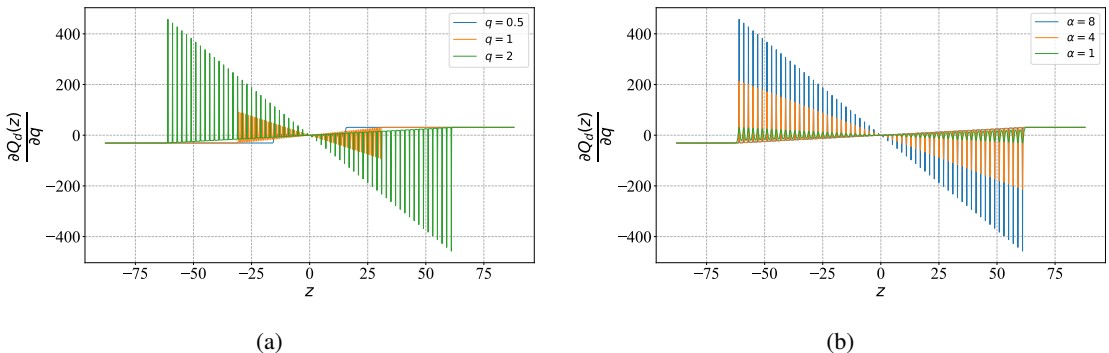

(a)                                                                          (b)

Figure 8: Figures 8a and **??** illustrate the partial derivatives of $Q_d(z)$ w.r.t. the parameter $q$ for cases where $\alpha = 8$ and $q$ varies, and $\alpha$ varies and $q = 2$, respectively. For both figures, we set $b = 6$.

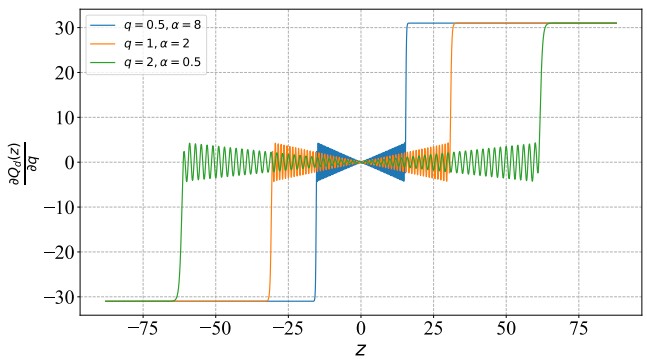

Figure 9: Demonstrating impact of $\hbar$ on gradient magnitude for various combinations of $q$ and $\alpha$. We fixed $\hbar=2$ and $b=6$.

### A.3   CNN-BASED ARCHITECTURES SETTING

For CIFAR100, we deploy a stochastic gradient descent (SGD) optimizer with a momentum of 0.9, a weight decay of 0.0005, and a batch size of 64. We initialize the learning rate as 0.05, and decay it by 0.1 every 30 epochs after the first 150 epochs until the last 240 epoch. For MobileNetV2, ShuffleNetV1 and ShuffleNetV2, we use a learning rate of 0.01 as this learning rate is optimal for these models in a grid search, while 0.05 is optimal for other models.

For fine-grained classification tasks, all networks are trained from scratch and optimized by SGD with a momentum of 0.9, weight decay of 0.0001, and an initial learning rate of 0.1,. The learning rate is divided by 10 after epochs 100 and 150 for all datasets, and the total epochs are 200. We set batch size 32 for these fine-grained classification tasks. We use the standard data augmentation technique for ImageNet (Deng et al., 2009), *i.e.*, flipping and random cropping. This experimental setup is also outlined by Yun et al. (2020).

### A.4 Fine-grained Model Architectures

We use standard ResNet-18 with 64 filters and DenseNet-121 with a growth rate of 32 for image size $224 \times 224$. For fine-grained classification tasks, we use PreAct ResNet-18 (He et al., 2016), which modifies the first convolutional layer with kernel size $3 \times 3$, strides 1 and padding 1, instead of the kernel size $7 \times 7$, strides 2 and padding 3, for image size $32 \times 32$ by following Zhang et al. (2017). We use DenseNet-BC structure (Huang et al., 2017), and the first convolution layer of the network is also modified in the same way as in PreAct ResNet-18 for image size $32 \times 32$.

### A.5 Transformer-based Settings and Results

In this section, we compare the performance of JPEG-DL compared to its baseline using the EfficientFormer-L1 model (Li et al., 2022) on CIFAR-100, as well as two fine-grained datasets, Flowers and Pets. We followed the experimental setup described by Xu et al. (2023), adhering to the same configurations mentioned in Table 4. The learning rate was set to 0.003 for CIFAR-100 and 0.005 for the fine-grained tasks, aligning with our standard settings in Section 4.

Interestingly, we found that the best performance for transformer-based architectures was achieved when the quantization tables were initialized with all ones, effectively representing the highest quality factor quantization table for JPEG. The top-1 validation accuracy performance is presented in Table 5, demonstrating the significant improvements achieved by JPEG-DL over the baseline.

Table 4: Hyper-parameter setting on EfficientFormer-L1.

| Settings | CIFAR-100 | Pets | Flowers |
|---|---|---|---|
| batch size | 512 | 512 | 512 |
| warmup epochs | 50 | 100 | 100 |
| training epochs | 300 | 600 | 600 |

Table 5: Top-1 accuracy (%) on CIFAR-100 and two fine-grained image classification tasks using EfficientFormer-L1. We report the mean and standard deviation of the experimental results over three runs with different random seeds. The best results are indicated in bold, and values in parentheses indicate relative accuracy gains over the baseline.

| Model | Method | Accuracy (%) |
|---|---|---|
| CIFAR-100 | Baseline | $80.27_{\pm0.33}$ |
| | JPEG-DL | $\mathbf{80.49_{\pm0.23}}$  (+0.22) |
| Flowers | Baseline | $69.78_{\pm0.33}$ |
| | JPEG-DL | $\mathbf{73.05_{\pm0.12}}$  (+3.27) |
| Pets | Baseline | $65.52_{\pm1.23}$ |
| | JPEG-DL | $\mathbf{69.28_{\pm0.42}}$  (+3.76) |

### A.6 Comparison with more Baselines

In Table 6, we have included comparisons against other baselines that employed JPEG quantization in the pipeline, but with heuristic approaches to handle the non-differentiability and zero derivative problems of JPEG quantization. For example, Ballé et al. (2016) allow optimization via stochastic gradient descent by replacing the quantizer with an additive i.i.d. uniform noise, which has the same width as the quantization

bins, where $\hat{z} = z + q * U(-0.5, 0.5)$. However, during validation, they employed $Q_u$ using the trained quantization tables. Shin & Song (2017) employ a third-order polynomial approximation of the rounding function to make JPEG differentiable. The gradient approximation through the round function is a key aspect of certain neural network approaches. Esser et al. (2019)[1] employ a straight-through estimator, originally proposed by Bengio et al. (2013), to achieve this approximation. This method treats the round function as a pass-through operation during backpropagation, allowing for effective gradient estimation. Our experimental results demonstrate that we consistently outperform all tested baselines. Even though these methods find a way to make the quantization differentiable, they still hurt the performance. This is consistent with the well-known wisdom in the literature, which also shows that they did not take advantage of the higher level of non-linearity introduced into DNN architectures, as proposed by our JPEG-DL framework.

Table 6: Top-1 validation accuracy (%) on various fine-grained image classification tasks and model architectures. We report the mean and standard deviation of experimental results over three runs.

| Model | Method | CUB-200 | Dogs | Flowers | Pets |
|---|---|---|---|---|---|
| ResNet-18 | Baseline | $54.00_{\pm 1.43}$ | $63.71_{\pm 0.32}$ | $57.13_{\pm 1.28}$ | $70.37_{\pm 0.84}$ |
| | Ballé et al. (2016) | $50.78_{\pm 2.21}$ (-3.22) | $53.47_{\pm 7.37}$ (-10.24) | $55.46_{\pm 0.59}$ (-1.67) | $56.14_{\pm 17.16}$ (-14.23) |
| | Shin & Song (2017) | $55.34_{\pm 0.14}$ (+1.34) | $63.03_{\pm 0.56}$ (-0.68) | $55.78_{\pm 1.44}$ (-1.35) | $71.45_{\pm 1.01}$ (+1.08) |
| | Esser et al. (2019) | $51.58_{\pm 0.18}$ (-2.42) | $60.45_{\pm 0.23}$ (-3.26) | $58.04_{\pm 0.58}$ (+0.91) | $68.81_{\pm 0.55}$ (-1.56) |
| | JPEG-DL | $\mathbf{58.81}_{\pm 0.12}$ (+4.81) | $\mathbf{65.57}_{\pm 0.37}$ (+1.86) | $\mathbf{68.76}_{\pm 0.57}$ (+11.63) | $\mathbf{74.84}_{\pm 0.66}$ (+4.47) |
| DenseNet-121 | Baseline | $57.70_{\pm 0.44}$ | $66.61_{\pm 0.17}$ | $51.32_{\pm 0.57}$ | $70.26_{\pm 0.79}$ |
| | Ballé et al. (2016) | $52.00_{\pm 1.41}$ (-5.70) | $60.07_{\pm 6.41}$ (-6.54) | $46.60_{\pm 2.87}$ (-4.72) | $61.91_{\pm 1.88}$ (-8.35) |
| | Shin & Song (2017) | $57.19_{\pm 0.78}$ (-0.51) | $66.90_{\pm 0.13}$ (+0.29) | $51.04_{\pm 0.87}$ (-0.28) | $69.95_{\pm 1.21}$ (-0.31) |
| | Esser et al. (2019) | $56.46_{\pm 0.30}$ (-1.24) | $64.89_{\pm 0.12}$ (-1.72) | $55.98_{\pm 0.24}$ (+4.60) | $69.58_{\pm 0.59}$ (-0.68) |
| | JPEG-DL | $\mathbf{61.32}_{\pm 0.43}$ (+3.62) | $\mathbf{69.67}_{\pm 0.58}$ (+3.06) | $\mathbf{72.22}_{\pm 1.05}$ (+20.90) | $\mathbf{75.90}_{\pm 0.68}$ (+5.64) |

### A.7 LAYER REPLACEMENT

We extend our evaluation of the JPEG-DL framework to include layer replacement, where we substitute the ReLU activation function with our JPEG layer in ResNet-18. This replacement is implemented directly after the first convolution layer, which has 64 kernels and outputs a feature map of 112x112. For this layer replacement, we did not perform any color space conversion and added a quantization table with a size of 8x8 for each kernel output. We follow the experimental setup mentioned in Section 4, except we use a learning rate of 0.001, $\alpha$ equal to 2, and $b$ equal to 6 bits. The performance of this approach is shown in the last row of the following table and is compared to the performance of JPEG-DL when the JPEG layer is placed directly after the input layer shown in Fig. 1. The results shown in Table 7 confirm that this new architecture, resulting from layer replacement with the JPEG layer or inserting the JPEG layer directly after the Input Layer, indeed improves deep learning performance by a significant margin, thereby showing the potential of new forms of nonlinear operation in DNN architectures.

### A.8 COMPARISON WITH JPEG-BASED DATA AUGMENTATION AND OTHER PREPROCESSING METHODS

**JPEG-based Data Augmentation.** We have compared and implemented JPEG-based data augmentation across three different sets, each with varying ranges of quantity factor (QF). For each tested range, we randomly select a QF for each image within the mini-batch. These sets have been tested on fine-grained datasets

---

[1] We followed the experimental setup defined by Esser et al. (2019) using Stochastic Gradient Descent (SGD) with an initial learning rate of 0.01, adhering to the same learning rate decay schedule as the underlying model. Additionally, we initialized $Q$ by setting it to $2|Q|/\sqrt{2^b - 1}$ and implemented gradient scaling during training mentioned in their paper. We also maintained consistency by using the same number of bits used by JPEG-DL.

Table 7: Top-1 validation accuracy (%) on various fine-grained image classification tasks and model architectures. We report the mean and standard deviation of experimental results over three runs.

| Method | CUB-200 | Dogs | Flowers | Pets |
|---|---|---|---|---|
| JPEG-DL (Input Layer) | $58.81_{\pm 0.12}$ (+4.81) | $65.57_{\pm 0.37}$ (+1.86) | $68.76_{\pm 0.57}$ (+11.63) | $74.84_{\pm 0.66}$ (+4.47) |
| JPEG-DL ($1^{st}$ Conv Layer) | $59.27_{\pm 0.04}$ (+5.27) | $65.33_{\pm 0.07}$ (+1.62) | $72.10_{\pm 1.46}$ (+14.97) | $76.11_{\pm 0.37}$ (+5.74) |

for all the models evaluated. Table 8 shows corresponding results for these tested sets. Consistent with the common knowledge in the literature, JPEG-based data augmentation in general degrades the accuracy performance.

**Non-Learnable and Learnable Preprocessing Methods.** We have tested the performance of non-learnable and learnable preprocessing methods during training and validation across all tested models on all tested fine-grained datasets, as demonstrated in Table 8. For non-learnable preprocessing, we have considered applying denoising using a Gaussian kernel and histogram equalization for each image within a mini-batch. As for the learnable one, we have compared it with Tu et al. (2023) as a learnable resize module, which has a bandpass nature in that it learns to boost details in certain frequency subbands that benefit the downstream recognition models. We have applied the same setup mentioned in their paper, in which they fine-tuned the model to achieve some improvement [2].

**Impact of chrominance subsampling on JPEG-DL.** In Table 8, we evaluate various subsampling schemes for all tested models across all tested fine-grained datasets. Notably, although there is no clear winner among different chroma subsampling methods, DNNs indeed respond to color information differently from human; color information is more important to DNNs than human since chroma subsampling formats 4:4:4 and 4:2:2, in general, give rise to better accuracy performance than the 4:2:0 subsampling format which is adopted predominantly in the image and video coding.

## A.9 FIXING HYPERPARAMTERS ACROSS DIFFERENT DATASETS

In Table 9, we evaluate the difference between the hyperparameters used for CIFAR-100 and the fine-grained task, where the only difference lies in the learning rate. In the following setup, we present the results when a learning rate of 0.003 is applied to the fine-grained dataset. We observe that the performance difference is marginal, and in some cases, we can even achieve higher gains in model performance. This suggests that the choice of learning rate can have a significant impact on the effectiveness of the model.

## A.10 SENSITIVITY AND Q TABLE INITIALIZATION

Figure 10 presents the estimated sensitivity values for all models considered in the CUB200 dataset and a subset of models used for CIFAR-100, as proposed by Salamah et al. (2024b); Zheng et al. (2023). These estimated sensitivity values are used to initialize the quantization table for JPEG-DL. Figure 11 illustrates the initial and converged values of the quantization table at the end of training.

Table 8: JPEG-DL is compared to JPEG-based augmentation (orange), non-learnable (yellow), and learnable preprocessing (blue) methods. JPEG-DL employs three different chrominance subsampling schemes shown in green color. Top-1 validation accuracy (%) on various fine-grained image classification tasks and model architectures. We report the mean and standard deviation of experimental results over three runs. The best results are indicated in bold, and values in parentheses indicate relative accuracy gains over the baseline.

| Dataset | Method | | ResNet-18 | | DenseNet-121 | |
|---|---|---|---|---|---|---|
| CUB-200 | Baseline | | $54.00_{\pm1.43}$ | | $57.70_{\pm0.44}$ | |
| | Rand. QF | [1:50] | $52.86_{\pm0.69}$ | (-1.14) | $55.72_{\pm0.72}$ | (-1.98) |
| | Rand. QF | [50:100] | $54.53_{\pm0.41}$ | (+0.53) | $56.98_{\pm0.90}$ | (-0.72) |
| | Rand. QF | [1:100] | $53.91_{\pm0.43}$ | (-0.09) | $57.20_{\pm0.33}$ | (-0.50) |
| | Denoising | | $54.82_{\pm0.40}$ | (+0.82) | $56.02_{\pm1.00}$ | (-1.68) |
| | Equalization | | $49.00_{\pm0.31}$ | (-5.00) | $54.32_{\pm0.57}$ | (-3.38) |
| | Learnable Resize | | $54.98_{\pm0.19}$ | (+0.98) | $57.35_{\pm0.22}$ | (-0.35) |
| | JPEG-DL (4:2:0) | | $58.00_{\pm0.12}$ | (+4.00) | $60.55_{\pm0.71}$ | (+2.85) |
| | JPEG-DL (4:2:2) | | $58.11_{\pm0.10}$ | (+4.11) | $\mathbf{61.51}_{\pm0.41}$ | (+3.81) |
| | JPEG-DL (4:4:4) | | $\mathbf{58.81}_{\pm0.12}$ | (+4.81) | $61.32_{\pm0.43}$ | (+3.62) |
| Dogs | Baseline | | $63.71_{\pm0.32}$ | | $66.61_{\pm0.17}$ | |
| | Rand. QF | [1:50] | $60.61_{\pm0.17}$ | (-3.10) | $64.93_{\pm0.17}$ | (-1.68) |
| | Rand. QF | [50:100] | $63.14_{\pm0.30}$ | (-0.57) | $66.97_{\pm0.18}$ | (+0.36) |
| | Rand. QF | [1:100] | $62.12_{\pm0.16}$ | (-1.59) | $65.59_{\pm0.54}$ | (-1.02) |
| | Denoising | | $62.52_{\pm0.41}$ | (-1.19) | $66.36_{\pm0.34}$ | (-0.25) |
| | Equalization | | $62.38_{\pm0.29}$ | (-1.33) | $67.12_{\pm0.19}$ | (+0.51) |
| | Learnable Resize | | $63.10_{\pm0.43}$ | (-0.61) | $67.90_{\pm0.32}$ | (+1.29) |
| | JPEG-DL (4:2:0) | | $65.57_{\pm0.31}$ | (+1.86) | $68.90_{\pm0.08}$ | (+2.29) |
| | JPEG-DL (4:2:2) | | $\mathbf{65.64}_{\pm0.16}$ | (+1.93) | $\mathbf{69.85}_{\pm0.78}$ | (+3.24) |
| | JPEG-DL (4:4:4) | | $65.57_{\pm0.37}$ | (+1.86) | $69.67_{\pm0.58}$ | (+3.06) |
| Flowers | Baseline | | $57.13_{\pm1.28}$ | | $51.32_{\pm0.57}$ | |
| | Rand. QF | [1:50] | $58.33_{\pm0.48}$ | (+1.20) | $51.44_{\pm0.79}$ | (+0.12) |
| | Rand. QF | [50:100] | $55.98_{\pm0.89}$ | (-1.15) | $51.67_{\pm1.10}$ | (+0.35) |
| | Rand. QF | [1:100] | $57.55_{\pm0.66}$ | (+0.42) | $52.32_{\pm0.73}$ | (+1.00) |
| | Denoising | | $57.28_{\pm1.06}$ | (+0.15) | $50.37_{\pm0.79}$ | (-0.95) |
| | Equalization | | $60.56_{\pm0.98}$ | (+3.43) | $61.72_{\pm0.21}$ | (+10.40) |
| | Learnable Resize | | $57.41_{\pm0.48}$ | (+0.28) | $51.21_{\pm0.82}$ | (-0.11) |
| | JPEG-DL (4:2:0) | | $67.58_{\pm1.50}$ | (+10.45) | $68.01_{\pm1.17}$ | (+16.69) |
| | JPEG-DL (4:2:2) | | $67.75_{\pm1.19}$ | (+10.62) | $68.79_{\pm1.08}$ | (+17.47) |
| | JPEG-DL (4:4:4) | | $\mathbf{68.76}_{\pm0.57}$ | (+11.63) | $\mathbf{72.22}_{\pm1.05}$ | (+20.90) |
| Pets | Baseline | | $70.37_{\pm0.84}$ | | $70.26_{\pm0.79}$ | |
| | Rand. QF | [1:50] | $69.71_{\pm0.54}$ | (-0.66) | $68.93_{\pm0.50}$ | (-1.33) |
| | Rand. QF | [50:100] | $70.00_{\pm0.57}$ | (-0.37) | $68.63_{\pm0.82}$ | (-1.63) |
| | Rand. QF | [1:100] | $69.52_{\pm0.62}$ | (-0.85) | $70.60_{\pm1.05}$ | (+0.34) |
| | Denoising | | $69.15_{\pm1.07}$ | (-1.22) | $69.03_{\pm1.13}$ | (-1.23) |
| | Equalization | | $73.00_{\pm0.79}$ | (+2.63) | $75.04_{\pm1.30}$ | (+4.78) |
| | Learnable Resize | | $70.04_{\pm0.72}$ | (-0.33) | $68.83_{\pm1.85}$ | (-1.43) |
| | JPEG-DL (4:2:0) | | $74.64_{\pm1.34}$ | (+4.27) | $\mathbf{76.21}_{\pm0.35}$ | (+5.95) |
| | JPEG-DL (4:2:2) | | $74.81_{\pm0.51}$ | (+4.44) | $76.17_{\pm1.83}$ | (+5.90) |
| | JPEG-DL (4:4:4) | | $\mathbf{74.84}_{\pm0.66}$ | (+4.47) | $75.90_{\pm0.68}$ | (+5.64) |

Table 9: Top-1 validation accuracy (%) on various fine-grained image classification tasks and model architectures. We report the mean and standard deviation of experimental results over three runs.

| Model | Learning Rate | CUB-200 | Dogs | Flowers | Pets |
|---|---|---|---|---|---|
| ResNet-18 | 0.005 | $\mathbf{58.81}_{\pm0.12}$ (+4.81) | $\mathbf{65.57}_{\pm0.37}$ (+1.86) | $68.76_{\pm0.57}$ (+11.63) | $74.84_{\pm0.66}$ (+4.47) |
| | 0.003 | $58.62_{\pm0.50}$ (+4.61) | $65.45_{\pm0.17}$ (+1.74) | $\mathbf{69.61}_{\pm1.16}$ (+12.48) | $\mathbf{74.90}_{\pm0.82}$ (+4.53) |
| DenseNet-121 | 0.005 | $\mathbf{61.32}_{\pm0.43}$ (+3.62) | $\mathbf{69.67}_{\pm0.58}$ (+3.06) | $72.22_{\pm1.05}$ (+20.90) | $\mathbf{75.90}_{\pm0.68}$ (+5.64) |
| | 0.003 | $60.92_{\pm0.50}$ (+3.22) | $69.53_{\pm0.49}$ (+2.92) | $\mathbf{72.45}_{\pm0.92}$ (+21.13) | $75.83_{\pm0.64}$ (+5.56) |

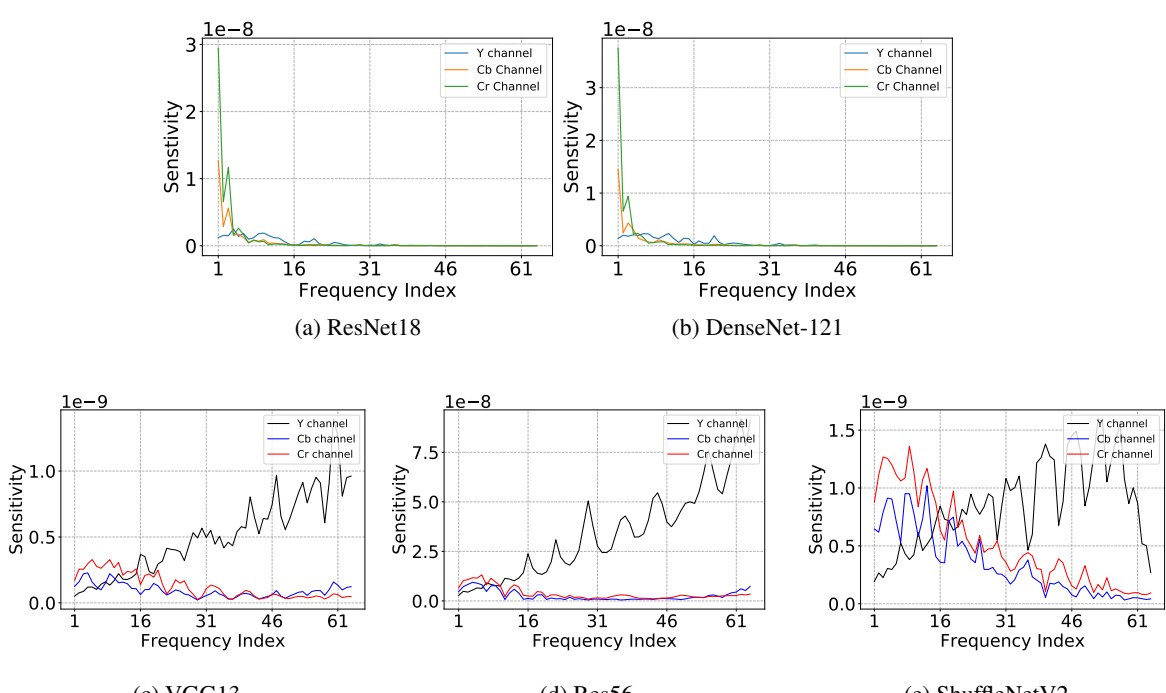

(a) ResNet18

(b) DenseNet-121

(c) VGG13

(d) Res56

(e) ShuffleNetV2

Figure 10: The estimated sensitivity is shown for two pre-trained models on the CUB200 (first row) and three pre-trained models on the CIFAR-100 (second row), with sensitivity indices arranged in the default zigzag order.

## A.11 FEATURE MAPS VISUALIZATION

Figure 12 presents the feature maps extracted after the first dense block in DenseNet-121 for both the JPEG-DL and the baseline models, trained on the Flowers dataset using the model from Table 2. The shown example was incorrectly classified by the baseline model, while the JPEG-DL model correctly classified it.

x

---

[2]We have verified the correctness of our PyTorch implementation with the authors' TensorFlow implementation provided in https://colab.research.google.com/github/google-research/google-research/blob/master/muller/muller_demo.ipynb.

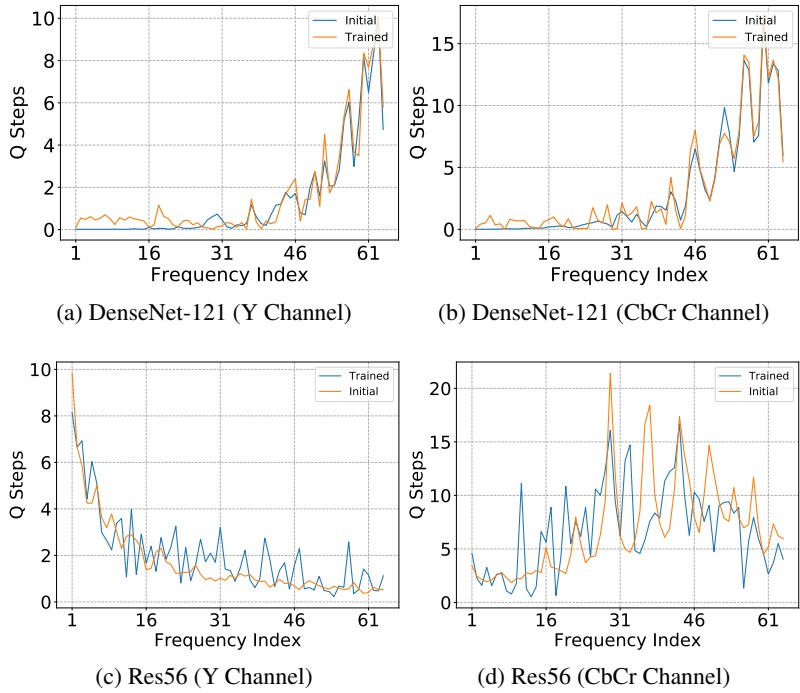

(a) DenseNet-121 (Y Channel)  (b) DenseNet-121 (CbCr Channel)

(c) Res56 (Y Channel)  (d) Res56 (CbCr Channel)

Figure 11: Initial and final quantization tables for Res56 trained on CIFAR-100 and DenseNet-121 trained on CUB200.

### A.12 MITIGATING THE INFERENCE OVERHEAD OF DIFFERENTIABLE SOFT QUANTIZERS

The reconstruction space $\hat{\mathcal{A}}$ is configured with a specific size, determined by the parameter $L$, which is set to $2^{b-1}$, as mentioned in Section 4. For CIFAR-100 and Fine-grained datasets, $b$ is set to 8 bits, resulting in a reconstruction space length of 513. For ImageNet-1K, $b$ is set to 11 bits, resulting in a length of 2047. This high dimensionality of the reconstruction space poses a computational challenge during inference when calculating the conditional probability mass function (CPMF) $P_\alpha(\cdot|z)$. To address this, the support of each CPMF is restricted to the five closest points in $\hat{\mathcal{A}}$ to the coefficient $z$ being quantized. This simplification, referred to as the masked CPMF, significantly reduces computational complexity without compromising the effectiveness of the quantization scheme $\mathcal{Q}_d$.

Figure 13 demonstrates the rapid decay of probability mass in CPMFs as the reconstruction level $\hat{z}$ deviates from the quantized value the uniform quantizer $\mathcal{Q}_u(z)$. The set $\mathcal{M}$, consisting of five reconstruction levels centered around $\mathcal{Q}_u(z)$, captures a significant portion of the total probability mass, even in extreme cases where the quantization step $q$ and the parameter $\alpha$ are very small. This observation suggests that restricting the CPMF support to $\mathcal{M}$ (masked CPMF) is a valid simplification. Instead of computing the full-space CPMF with length $2L + 1$, we can efficiently calculate the masked CPMF with length 5 using softmax on $\mathcal{M}$. The impact of this simplification on the quantization scheme $\mathcal{Q}_d$ is negligible, particularly for high values of $\alpha$, as shown in Table 10. Therefore, using the masked CPMF in our implementation is a justified simplification that significantly reduces computational complexity without compromising accuracy.

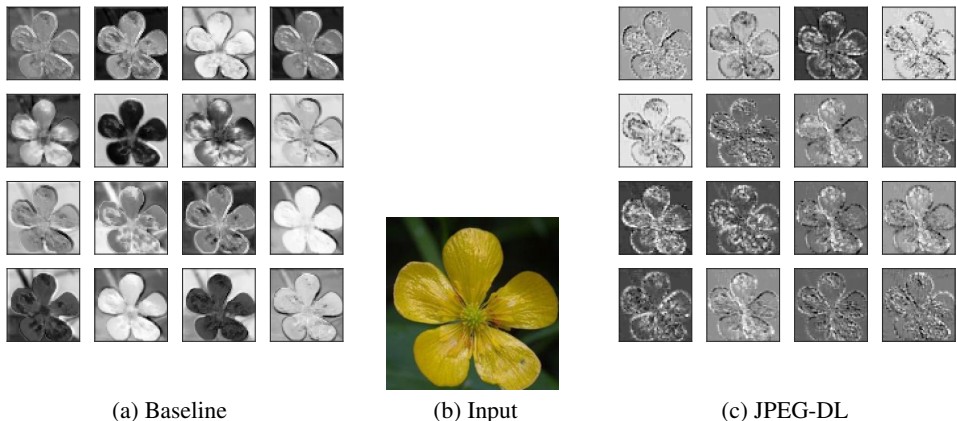

|        (a) Baseline        |        (b) Input        |        (c) JPEG-DL        |

Figure 12: Feature maps of size 56×56 are shown after the first dense block in DenseNet-121 for both JPEG-DL and baseline models in Figs. 12a and 12c, respectively, using an original input shown in Fig. 12b. The JPEG-DL model highlights the foreground (flower) more distinctly, while the baseline model shows less contrast, contributing to its misclassification.

| $\alpha$ | \multicolumn{3}{c}{$\mathcal{Q}_d(z = 0.5)$} |
| --- | --- | --- | --- |
|  | Full Space | Masked | $\Delta$ |
| 1 | 0.5 | 0.5027 | 0.0027 |
| 3 | 0.5 | 0.5 | 5.96e-08 |
| 5 | 0.5 | 0.5 | 5.96e-08 |
| 10 | 0.5 | 0.5 | 0 |

Table 10: $\mathcal{Q}_d(z = 0.5; q = 1, \alpha)$ values resulting from the full-space CPMF and the masked CPMF at different $\alpha$, alongside the absolute difference $\Delta$ between the full-space and masked cases.

| Inference Settings | Inference time (milliseconds) | Inference Throughput (img/sec) |
| --- | --- | --- |
| Baseline | 5.46 | 180.65 |
| JPEG-DL (Masked CPMFs) | 7.43 | 133.79 |
| JPEG-DL (Full Space) | 18.86 | 52.37 |

Table 11: Comparison of inference time between different inference settings measured for Resnet18 on ImageNet, following the experimental setup mentioned in Section 4. Evaluation is conducted on a single NVIDIA RTX A5000 GPU.

Specifically, Fig. 13 illustrates how fast the probability mass of a reconstruction level $\hat{z}$ decays in CPMFs as $\hat{z}$ deviates from the uniform quantizer $\mathcal{Q}_u(z)$. It's clear that the set $\mathcal{M} = \{\mathcal{Q}_u(z) - 2q, \mathcal{Q}_u(z) - q, \mathcal{Q}_u(z), \mathcal{Q}_u(z) + q, \mathcal{Q}_u(z) + 2q\}$ of reconstruction levels retain a total probability mass close to 1, even

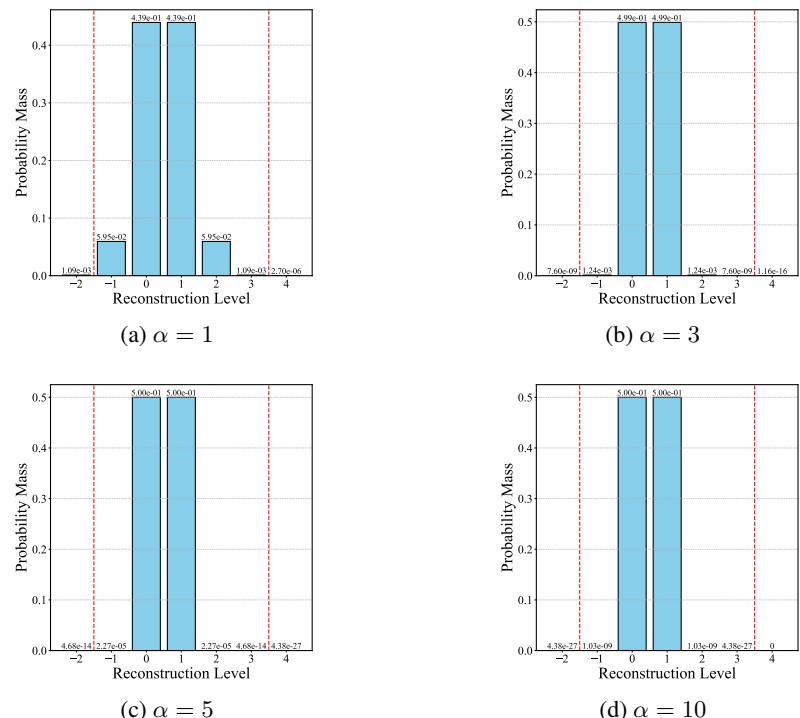

Figure 13: Partial visualization of CPMFs $P_\alpha(\cdot|z = 0.5)$ computed using (6), with $L = 1023$, $q = 1$, and $\alpha$ values of (a) 1, (b) 3, (c) 5, and (d) 10. The reconstruction levels between two red dashed lines represent $\mathcal{M}$.

in an adversarially chosen scenario where both $q$ and $\alpha$ are extremely small[3] (i.e., 1) and the coefficient $C$ being quantized lies exactly at a quantization threshold. Therefore, instead of computing the full-space CPMF with length $2L + 1$ following (6), we only conduct softmax on $\mathcal{M}$ to get the masked CPMF with length 5. The error in $\mathcal{Q}_d$ caused by this masking is negligible as shown in Table 10, especially in the high $\alpha$ case. Therefore, it's totally justified to simplify the full-space CPMF to the masked CPMF in our implementation.

Table 11 presents a comparison of inference time and throughput between the standard model and our unified model, which incorporates the JPEG layer with the underlying model. The inference times for both the full-space CPMF and the masked CPMF are measured. The reported inference time is averaged across the entire validation set of ImageNet using Resnet18 from Table 3. These results demonstrate the effectiveness of the proposed approach in addressing the computational complexity of the JPEG-DL framework.

---

[3]As $q$ and $\alpha$ increase, the CPMF becomes even sharper, and as a result, $\mathcal{M}$ will capture increasingly higher probability mass.

