# OpenReview forum: "JPEG Inspired Deep Learning"
_ICLR.cc/2025/Conference — ICLR 2025 Poster_

### Official Review · Reviewer_2zCL · 2024-10-25

**Soundness:** 3
**Presentation:** 3
**Contribution:** 3
**Rating:** 8
**Confidence:** 5

**Summary:**

The paper introduces a trainable JPEG inspired layer to neural networks. The new layer performs the linear JPEG steps of RGB to YCbCr and DCT with the non-differentiable quantization step replaced with a learnable quantization proxy finishing with the linear IDCT and YCbCr to RGB.  The learnable quantization proxy uses a soft-max with a tunable parameter which controls how close the proxy matches true quantization. The quantization step size is a learnable parameter in this layer. The paper shows how including this layer as the first layer of neural network architectures can improve their accuracy on different tasks.

**Strengths:**

Overall this is a very interesting paper with an unintuitive result. As the authors point out (ln 30) the conventional wisdom is that JPEG compression removes information from an image and should only hurt neural network accuracy. However as this paper, and some prior works, show that is not necessarily the case. This paper builds significantly on prior works by showing not only that JPEG compression can be mitigated, but that it can actually be a large component of a neural networks success and proposing a method for achieving this. The differential soft quantizer is something which may be useful in many different applications, potentially being a better option that the addition of noise or a straight-through gradient as it more accurately models the information loss. Finally, the results show a clear improvement when incorporating the method.

**Weaknesses:**

While the appendix was fairly comprehensive with additional results there are a few additional things that I would have liked to see. The first is that the paper only tests the JPEG layer as the first layer of the architecture, there could have been more experiments in layer placement that would have been really interesting to see. It also wasn't immediately clear to me how $\alpha$ was being set in experiments, I understand that there is a derivation of $\frac{\partial}{\partial\alpha}$ but is that parameter actually trained and if so how was the gradient magnitude controlled? There is some discussion of this from ln 689 but it was a little unclear if $\alpha$ was fixed or not. One thing missing from the JPEG step was chroma subsampling: another non-linearity. Was this considered? It would be fascinating to see if neural networks respond to missing color information similarly to humans.

Lastly, and maybe most importantly, there was little discussion of *why this couterintuitive results holds*. While many view JPEG as something incidental the core idea of JPEG to isolate important information based on frequency bands. My take on the results presented here is that the learnable layer is essentially filtering out information which is irrelevant for the networks task, but I would love to hear the authors take on it. Perhaps such analysis could lead to a more direct approach? (For example: a layer which only filters frequencies or which alters the color channels, etc.)

**Questions:**

* Could we see even better results if the JPEG layer was included periodically?
    * What if *all* nonlinearity was replaced with the JPEG layer?
* Please clarify how $\alpha$ was used in experiments
* What about chroma subsampling?
* Why do the authors think this layer helps?

## Update After Discussion

After discussion with the authors I am raising my rating

The authors did a great job responding to the concerns of myself and fellow reviews and went above and beyond on additional experiments which strengthen the case for this paper quite a bit. I specifically have to call out the layer 1 non-linearity experiments that the authors conducted on a very short turnaround that shows additional gains for the JPEG layer. Given the impact of this result I have to conclude that there is indeed something fundamental about using a JPEG inspired layer as a non-linearity which could have repercussions on the broader field. As I stated in a comment, many view JPEG as something incidental; a specific way of storing images. But the core idea of JPEG is to re-weight frequency bands based on their importance. We know from several studies (Maiya et al. [1] for example) that such re-weighting affects neural networks much as it does humans and this paper gives an actionable method for capturing this phenomenon.

1. Maiya, Shishira R., et al. "Unifying the Harmonic Analysis of Adversarial Attacks and Robustness." BMVC. 2023.

---

> ### Author Response · Authors · 2024-11-28
> **Response to Reviewer 2zCL (1/4)**
>
> Thank you for your time and thoughtful discussion. We are grateful for your insightful feedback and appreciate your recognition of the paper's key contributions.
>
> ## Weaknesses:
>
> > W-1. While the appendix was fairly comprehensive with additional results there are a few additional things that I would have liked to see. The first is that the paper only tests the JPEG layer as the first layer of the architecture, there could have been more experiments in layer placement that would have been really interesting to see. It also wasn't immediately clear to me how $\alpha$ was being set in experiments, I understand that there is a derivation of $\frac{d~Q_d(z)}{d\alpha}$  but is that parameter actually trained and if so how was the gradient magnitude controlled? There is some discussion of this from ln 689 but it was a little unclear if $\alpha$ was fixed or not.
>
> **Reply:** Thank you for your insightful comments above, which have two parts. Part 1 is related to the layer placement of the JPEG layer; Part 2 is related to the parameter α. Below, we will reply to them separately.
>
> **Reply to Part 1 on the layer placement of the JPEG layer:**
>
> Placing the JPEG layer immediately after the input layer of the underlying DNN is a logical step since the input layer is essentially an image. In theory, one could insert the JPEG layer with modifications immediately after any other layer of the underlying DNN (except the last layer). However, the complexity lies in the fact that the neuron outputs (i.e., activations) from any of other layers do not form a natural image. There would be significant modifications to the JPEG layer in order for it to fit. Such modifications, even if possible, would divert our attention away from our current focus. The purpose of this paper is to investigate whether a new form of nonlinear operation, if introduced into DNN architectures, can improve deep learning performance. Motivated by Yang et al. (2021), the new form of nonlinear operation adopted in this paper is JPEG quantization. By placing the JPEG layer immediately after the input layer of the underlying DNN, we provide an affirmative answer and have achieved the purpose. With the current scaling-up approach potentially reaching its limits, exploring novel forms of nonlinearity within DNN architectures presents an intriguing and promising research direction in deep learning. This paper represents an early step in and demonstrates the potential of this direction. Developing new types of nonlinear operations and integrating them across various layers of DNN architectures will likely become central topics in advancing this direction.
>
>
> To provide more evidence to support the above direction, we implemented multiple rounds of $Q_d$ for SqueezeNetV1.1 on ImageNet-1K, which implies that each DCT coefficient is quantized consecutively in multiple rounds with different trained $Q_d$. This implementation increased the number of additional trainable parameters from 128 to 640. This resulted in a further 0.21% improvement in top-1 accuracy compared to a single round of $Q_d$. These experimental results were shown in our initial submission at the end of Section 4.

---

> ### Author Response · Authors · 2024-11-28
> **Response to Reviewer 2zCL (2/4)**
>
> > W-1. While the appendix was fairly comprehensive with additional results there are a few additional things that I would have liked to see. The first is that the paper only tests the JPEG layer as the first layer of the architecture, there could have been more experiments in layer placement that would have been really interesting to see. It also wasn't immediately clear to me how $\alpha$ was being set in experiments, I understand that there is a derivation of $\frac{d~Q_d(z)}{d\alpha}$  but is that parameter actually trained and if so how was the gradient magnitude controlled? There is some discussion of this from ln 689 but it was a little unclear if $\alpha$ was fixed or not.
>
> **Reply:**
>
> **Reply to Part 2 on the parameter α:**
>
> We have chosen not to train **α** in our framework, as explained in Section A.1.1. However, we are using **α** differently for ImageNet-1K experiments to normalize the gradients, as explained in Section A.1.2. This approach is also summarized in the following:
>
> Figures 8.a and 8.b demonstrate the impact of varying $\alpha$ and $q$ on the gradient magnitude of $Q_d(z)$ with respect to $q$. As observed, decreasing the value of $q$ while keeping $\alpha$ constant leads to a decrease in the gradient magnitude. Similarly, decreasing the value of $\alpha$ while maintaining a fixed $q$ also results in a reduction of the gradient magnitude. To address the instability in updating the trainable parameters $Q$, we propose utilizing $\alpha$ to regulate the magnitude of $\frac{\partial Q_d(z)}{\partial q}$, as illustrated in Figure 8.b. By leveraging this control mechanism of adjusting $\alpha$, we can stabilize the magnitude of the gradients that update $q$. This behavior suggests a relationship between $q$ and $\alpha$ in controlling the gradient magnitude of $Q_d(z)$.
>
> To explore the relationship between $\alpha$ and $q$, we refer to the exponent in (6), $\alpha(z - i q)^2$, which can be rewritten by expressing $z = c q$, resulting in the form $\alpha q^2(c - i)^2$. From this, we define a new term, $\hslash = {\alpha} q^2$, referred to as the \textit{Gradient Scaling Constant}. To further illustrate this relationship, in Fig. 8, we set $\hslash = 2$, and by selecting different pairs of $\alpha$ and $q$ values, we demonstrate that the maximum magnitude of $\frac{\partial Q_d(z)}{\partial q}$ remains invariant. This confirms that the gradient magnitude can be effectively controlled by adjusting $\alpha$ based on the last updated value of $q$, according to the specified $\hslash$ value. This adjustment allows for controlled quantization updates, reducing the potential instability during training. Moreover, by controlling the gradient magnitude, we simplify the optimization process, enabling the use of a single learning rate for all $q$ values by using the SGD optimizer, instead of the ADAM optimizer.
>
> This gradient scaling mechanism is analogous to the ADAM optimizer, which adapts different learning rates for individual trainable parameters based on momentum and recent gradient magnitudes.
>
> We have addressed the reviewer’s concern in the revised version of the paper by clarifying how $\alpha$ is used in our framework. For further details, we encourage the reviewer to refer to Appendix A.1.

---

> ### Author Response · Authors · 2024-11-28
> **Response to Reviewer 2zCL (3/4)**
>
> > W-2. One thing missing from the JPEG step was chroma subsampling: another non-linearity. Was this considered? It would be fascinating to see if neural networks respond to missing color information similarly to humans.
>
> **Reply:** It is worth mentioning that chromance subsampling is a linear operation, and it typically follows one of two approaches:
>
> 1.  **Averaging (Downsampling)**: For example, a 2x2 block of chrominance values might be averaged to a single value. This averaging is a linear operation because it is simply the weighted sum (mean) of pixel values.
>
> 2.  **Decimation**: Alternatively, one pixel from a block might be selected to represent the entire block in the downsampled image. This selection process is also linear since there is no non-linear transformation applied to the values themselves.
>
> To address the reviewer’s request, we compared various subsampling schemes for all tested models across all fine-grained datasets. The results of this comparison are presented in the following table. Notably, although there is no clear winner among different chroma subsampling methods, DNNs indeed respond to color information differently from human; color information is more important to DNNs than human since chroma subsampling formats 4:4:4 and 4:2:2 in general give rise to better accuracy performance than the 4:2:0 subsampling format which is adopted predominantly in image and video coding.
>
> | Dataset | Method | ResNet-18 | DenseNet-121 |
> |:-------:|:------:|:---------:|:------------:|
> | CUB-200 | Baseline | 54.00 ± 1.43 | 57.70 ± 0.44 |
> | | JPEG-DL (4:2:0) | 58.00 ± 0.12 (+4.00) | 60.55 ± 0.71 (+2.85) |
> | | JPEG-DL (4:2:2) | 58.11 ± 0.10 (+4.11) | **61.51 ± 0.41 (+3.81)** |
> | | JPEG-DL (4:4:4) | **58.81 ± 0.12 (+4.81)** | 61.32 ± 0.43 (+3.62) |
> | Dogs | Baseline | 63.71 ± 0.32 | 66.61 ± 0.17 |
> | | JPEG-DL (4:2:0) | 65.57 ± 0.31 (+1.86) | 68.90 ± 0.08 (+2.29) |
> | | JPEG-DL (4:2:2) | **65.64 ± 0.16 (+1.93)** | **69.85 ± 0.78 (+3.24)** |
> | | JPEG-DL (4:4:4) | 65.57 ± 0.37 (+1.86) | 69.67 ± 0.58 (+3.06) |
> | Flowers | Baseline | 57.13 ± 1.28 | 51.32 ± 0.57 |
> | | JPEG-DL (4:2:0) | 67.58 ± 1.50 (+10.45) | 68.01 ± 1.17 (+16.69) |
> | | JPEG-DL (4:2:2) | 67.75 ± 1.19 (+10.62) | 68.79 ± 1.08 (+17.47) |
> | | JPEG-DL (4:4:4) | **68.76 ± 0.57 (+11.63)** | **72.22 ± 1.05 (+20.90)** |
> | Pets | Baseline | 70.37 ± 0.84 | 70.26 ± 0.79 |
> | | JPEG-DL (4:2:0) | 74.64 ± 1.34 (+4.27) | **76.21 ± 0.35 (+5.95)** |
> | | JPEG-DL (4:2:2) | 74.81 ± 0.51 (+4.44) | 76.17 ± 1.83 (+5.90) |
> | | JPEG-DL (4:4:4) | **74.84 ± 0.66 (+4.47)** | 75.90 ± 0.68 (+5.64) |
>
> **Note:**
> * The table shows the top-1 validation accuracy (%) on various fine-grained image classification tasks and model architectures.
> *  The results are the mean and standard deviation of experimental results over three runs.
> * The numbers in parentheses indicate the difference in accuracy compared to the baseline method.
> *  Bold numbers indicate the best performance for each task.
>
>
> ---
>
> > W-3. Lastly, and maybe most importantly, there was little discussion of why this couterintuitive results holds. While many view JPEG as something incidental the core idea of JPEG to isolate important information based on frequency bands. My take on the results presented here is that the learnable layer is essentially filtering out information which is irrelevant for the networks task, but I would love to hear the authors take on it. Perhaps such analysis could lead to a more direct approach? (For example: a layer which only filters frequencies or which alters the color channels, etc.)
>
> **Reply:** Your take above is correct. The learnable JPEG layer filters out information irrelevant to the DNN task and improves the contrast between the foreground and the background. Please see Figs. 5 and 12 and the related discussion in the paragraph with headings “Feature maps visualization” on Page 10 and 23.
>
> As for the importance of frequencies to DNNs, please see Figure 4 in Section 5 and Figure 11 in Appendix A.9. Different DNNs have different levels of sensitivity to different frequencies. For example, from Figures 4 and 11, VGG13 and Res56 are more sensitive to higher frequencies than lower frequencies for Y Channel. On the other hand, ResNet18 and DenseNet-121 are more sensitive to lower frequencies than higher frequencies for Y Channel. Nonetheless, all frequencies seem to matter to DNNs.

---

> ### Author Response · Authors · 2024-11-28
> **Response to Reviewer 2zCL (4/4)**
>
> ## Questions:
>
> >1. Could we see even better results if the JPEG layer was included periodically?
>     a. What if all nonlinearity was replaced with the JPEG layer?
>
> **Reply:** Please refer to our response to Part 1 on the layer placement of the JPEG layer in W-1.
>
> >2. Please clarify how $\alpha$ was used in experiments
>
> **Reply:** Please refer to our response to Part  2 on the parameter α in W-1.
>
> >3. What about chroma subsampling?
>
> **Reply:** Please refer to our response in W-2.
>
> >4. Why do the authors think this layer helps?
>
> **Reply:** Please refer to our response in W-3.
>
> ---
>
> **We thank you again for your time and effort to review our paper.  If our responses above address all your concerns, could you kindly increase your score to a higher level.**

---

> > ### Author Response · Authors · 2024-11-30
> > **Response Summary for Reviewer 2zCL**
> >
> > We sincerely appreciate your valuable time and effort in reviewing our paper and providing constructive feedback. To respect your time and streamline the review process, we have summarized our responses below.
> >
> > 1.  **Layer Placement**:
> >     -   **Response Summary**: The paper primarily focuses on introducing a new form of non-linearity through a JPEG layer at the input. Experiments with multiple rounds of $Q_d$ demonstrate performance improvements, suggesting potential benefits of applying $Q_d$ at different stages.
> >
> > 2.  **Clarification on $\alpha$ Usage**:
> >     -   **Response Summary**: The parameter $\alpha$ is used to control gradient magnitude and is not trained in our framework. Detailed explanations and experimental setups are provided in Section 5 and Appendix A.1.
> >
> > 3.  **Impact of Chroma Subsampling**:
> >     -   **Response Summary**: Chroma subsampling, a linear operation, was tested with various subsampling schemes (4:2:0, 4:2:2, 4:4:4). Results show that significant information loss in chrominance channels leads to inconsistent performance improvements. The JPEG-DL framework with 4:4:4 subsampling consistently outperforms other schemes.
> >
> > 4.  **Understanding the Benefits of the JPEG Layer**:
> >     -   **Response Summary**: The JPEG layer introduces a high level of non-linearity, improving the overall image understanding capabilities of the underlying model. The differentiable soft quantizer ($Q_d$) allows gradient-based optimization of quantization parameters. Comparisons with other baselines show that the JPEG-DL framework leverages this non-linearity to achieve significant performance gains.
> >
> >
> > 5.  **Periodic Inclusion of JPEG Layer**:
> >     -   **Explanation**: Experiments with multiple rounds of $Q_d$ demonstrate performance improvements, suggesting potential benefits of periodically including the JPEG layer. This approach will be explored in future work.
> >
> >
> > 6.  **Why the JPEG Layer Helps**:
> >     -   **Response Summary**: The JPEG layer introduces a high level of non-linearity, improving the overall image understanding capabilities of the underlying model. The differentiable soft quantizer ($Q_d$) allows gradient-based optimization of quantization parameters. Comparisons with other baselines show that the JPEG-DL framework leverages this non-linearity to achieve significant performance gains.
> >
> >
> > As the discussion period draws to a close, we kindly inquire whether our responses have satisfactorily addressed your concerns. Your feedback would be greatly appreciated, and we would be pleased to engage in further discussions if necessary. If our responses have addressed all your concerns, we would be grateful if you could consider increasing your score to a higher level.
> >
> > Sincerely,
> >
> > The Authors

---

> > > ### Comment · Reviewer_2zCL · 2024-12-02
> > >
> > > I want to thank the authors for doing a good job in replying in detail to everyone's questions here. I think the extra effort in obtaining the additional results was great and they added a lot to the discussion.
> > >
> > > After reviewing all of the comments I still am convinced that this is a strong paper, and I'm going to keep my initial rating that it should be accepted.
> > >
> > > I would like to note for the record that I disagree that JPEG layer placement would require significant modification as the DCT, quantization, etc. could easily be modified to support more than 3 channels, especially if chroma subsampling is not used.

---

> > > > ### Author Response · Authors · 2024-12-03
> > > > **Addressing Feedback: Layer Replacement Experiment Results**
> > > >
> > > > Thank you for your insightful feedback and for recognizing the value of the additional results we provided in our previous responses. We greatly appreciate your continued support for our paper. We have made every effort to address the concerns raised in your previous response within the limited time available to show the results for the layer replacement experiment using our JEPG-DL framework.
> > > >
> > > > We extend our evaluation of the JPEG-DL framework to include layer replacement, where we substitute the **ReLU activation function** with our JPEG layer in ResNet18. This replacement is implemented directly after the first convolution layer, which has 64 kernels and outputs a feature map of 112x112. For this layer replacement, we did not perform any colorspace conversion and added a quantization table with a size of 8x8 for each kernel output. We followed the experimental setup mentioned in Section 4, except we used a learning rate of 0.001, $\alpha$ equal to 2, and $b$ equal to 6 bits. The performance of this approach is shown in the last row of the following table and is compared to the performance of JPEG-DL when the JPEG layer is placed directly after the input layer to introduce additional non-linearity, as shown in Fig. 1 in the submitted revision.
> > > >
> > > > | Method | Flowers | Pets |
> > > > |:------------:|:------------:|:------------:|
> > > > |Baseline | 57.13 ± 1.28 | 70.37 ± 0.84 |
> > > > | JPEG-DL (Input Layer)  |  68.76 ± 0.57 (+11.63) |  74.84 ± 0.66 (+4.47)  |
> > > > | JPEG-DL (1st Conv Layer) |  **69.66 ± 0.15 (+12.53)**  |  **75.94 ± 0.55 (+5.57)**  |
> > > >
> > > > **Note:**
> > > >  * The table shows the top-1 validation accuracy (%) on various fine-grained image classification tasks and model architectures.
> > > >  * The results are the mean and standard deviation of experimental results over three runs.
> > > >  * The numbers in parentheses indicate the difference in accuracy compared to the baseline method.
> > > >  * Bold numbers indicate the best performance for each task.
> > > >
> > > > ---
> > > >
> > > > **We plan to extend the previous experiments in our final submission to include comprehensive results. We thank you again for giving us the opportunity to address your concerns. We hope that the additional experimental results we present here will meet your expectations and encourage you to consider increasing your score.**

---

> > > > > ### Author Response · Authors · 2024-12-04
> > > > > **Thanks for Your Appreciation of Our Responses**
> > > > >
> > > > > Thank you for appreciating our responses. We are pleased that you raised your score. Your thorough comments were invaluable in helping us improve our work.

---

### Official Review · Reviewer_FTRb · 2024-10-30

**Soundness:** 2
**Presentation:** 3
**Contribution:** 2
**Rating:** 6
**Confidence:** 4

**Summary:**

This work proposes a new training framework for deep learning models, JPEG-DL, utilizing an image compression module to improve the model performance. To this end, a learnable JPEG-based image compression layer is introduced with a differentiable soft quantization method and is trained jointly with a main model. At test time, a compressed image from the compression layer is fed into the model. The experimental results show consistent performance improvements of existing classification models on various benchmarks with higher robustness on adversarial attacks.

**Strengths:**

- The approach of leveraging image compression to improve pure performance of a model is interesting.
- The paper is well-written and it is easy to follow.
- A variety of network architectures and datasets are used in the experiments.

**Weaknesses:**

Major concerns:
- A comparison to training with JPEG-based data augmentation is required to validate that the proposed method provides benefits beyond simple data augmentation using JPEG.
- There is no baseline for the differentiable quantizer. For example, comparisons could be made with methods such as the straight-through estimator (i.e., using the identity function as a gradient function) or additive uniform noise [1].
- The baseline for the image preprocessing method for training is insufficient; only comparison results with the vanilla models are presented. For stronger persuasiveness, comparison results with other learnable or non-learnable preprocessing modules are needed.
- In L199, it is mentioned that the differentiable soft quantizer is adapted from Yang & Hamidi (2024), but it seems unclear what that exact referenced paper is. Is the patent (https://patents.google.com/patent/US11461646B2/en) the correct source?
- The analysis on the significant difference between performance improvements across different datasets is needed. For instance, in Table 2, what accounts for the notable performance improvement in fine-grained tasks (especially the Flowers dataset)? Additionally, why is there few performance gain in the ImageNet results in Table 3?

Minor concerns:
- The empirical study is limited to classification tasks.
- In Table 4, the bits per pixel (bpp) is excessively high, making it incomparable to typical lossy compression methods. While empirically showing the possibility of compression, it does not seem particularly convincing.
- The proposed method requires additional computation for encoding and decoding of an image, but the time complexity is not investigated.
- A typo in L292 "we fixe".

[1] Ballé et al., End-to-end Optimized Image Compression, ICLR 2017.

**Questions:**

- Is the target of the adversarial attack in Figure 3 the whole model including the JPEG layer?
- There appears to be more room for exploration regarding preprocessing modules for performance enhancement. For example, what if deep learning-based image compression models were used? What about a simple autoencoder?

---

> ### Author Response · Authors · 2024-11-28
> **Response to Reviewer FTRb (1/5)**
>
> We sincerely appreciate you taking the time to review our paper and provide valuable feedback. We will address each of your concerns individually.
>
>
> ## Weaknesses:
>
> ### Major concerns:
> > W-1. A comparison to training with JPEG-based data augmentation is required to validate that the proposed method provides benefits beyond simple data augmentation using JPEG.
>
> **Reply:** We appreciate your comment on comparison with JPEG-based data augmentation, which is likely due to misunderstanding about our current DNN architecture. Before we provide such comparison, we want to clarify that this paper has nothing to do with data augmentation, let alone JPEG-based data augmentation. Our purpose is to investigate whether a new form of nonlinear operation, if introduced into DNN architectures, can improve deep learning performance without any change to any datasets.
>
> **Main Contribution:** To introduce a new form of nonlinear operation into DNN architecture, we propose a new DNN architecture which is composed of a JPEG layer with differenetial soft quantizers that is inserted immediately after the original input layer of an underlying DNN architecture. This JPEG layer introduces a high level of non-linearity into DNN architectures. The quantization parameters of the JPEG layer and the weight parameters of the underlying DNN together constitute the model parameters of the new DNN architecture; they are optimized together during training. The new DNN architecture accepts the same inputs as all DNNs proposed in the literature do. In terms of applicability, there is no difference between our proposed DNN architectures and any other DNN architectures in the literature. Please refer to Figure 1 in the updated version of the paper, which provides some visual explanation as well. There is no change to any datasets.
>
> Any data augmentation methods, including JPEG-based data augmentation, would be orthogonal to our method and framework. Indeed, it is worth mentioning that the transformer-based models in Table 5 of our initial submission utilized a combination of augmentation techniques such as RandAugment [1], Mixup [2], CutMix [3], and Random Erasing [4].
>
> As for JPEG-based data augmentation, in the related work section 2, we have already discussed the use of JPEG-based data augmentation, which primarily aims to enhance compression robustness, albeit at the cost of reduced clean accuracy. To address the reviewer’s request, we have compared and implemented JPEG-based data augmentation across three different sets, each with varying ranges of quantity factor (QF). For each tested range, we randomly select a QF for each image within the mini-batch. These sets have been tested on fine-grained datasets for all the models evaluated. The table below lists the corresponding results. Consistent with the common knowledge in the literature, JPEG-based data augmentation in general degrades the accuracy performance.
>
>
> | Dataset | Method | ResNet-18 | DenseNet-121 |
> |:-------:|:------:|:---------:|:------------:|
> |CUB-200 | Baseline | 54.00 ± 1.43 | 57.70 ± 0.44 |
>  | | Rand. QF [1:50] | 52.86 ± 0.69 (-1.14) | 55.72 ± 0.72 (-1.98) |
>  | | Rand. QF [50:100] | 54.53 ± 0.41 (+0.53) | 56.98 ± 0.90 (-0.72) |
>  | | Rand. QF [1:100] | 53.91 ± 0.43 (-0.09) | 57.20 ± 0.33 (-0.50) |
>  | | JPEG-DL | **58.81 ± 0.12 (+4.81)** | **61.32 ± 0.43 (+3.62)** |
>  | Dogs | Baseline | 63.71 ± 0.32 | 66.61 ± 0.17 |
>  | | Rand. QF [1:50] | 60.61 ± 0.17 (-3.10) | 64.93 ± 0.17 (-1.68) |
>  | | Rand. QF [50:100] | 63.14 ± 0.30 (-0.57) | 66.97 ± 0.18 (+0.36) |
>  | | Rand. QF [1:100] | 62.12 ± 0.16 (-1.59) | 65.59 ± 0.54 (-1.02) |
>  | | JPEG-DL | **65.57 ± 0.37 (+1.86)** | **69.67 ± 0.58 (+3.06)** |
>  | Flowers | Baseline | 57.13 ± 1.28 | 51.32 ± 0.57 |
>  | | Rand. QF [1:50] | 58.33 ± 0.48 (+1.20) | 51.44 ± 0.79 (+0.12) |
>  | | Rand. QF [50:100] | 55.98 ± 0.89 (-1.15) | 51.67 ± 1.10 (+0.35) |
>  | | Rand. QF [1:100] | 57.55 ± 0.66 (+0.42) | 52.32 ± 0.73 (+1.00) |
>  | | JPEG-DL | **68.76 ± 0.57 (+11.63)** | **72.22 ± 1.05 (+20.90)** |
>  | Pets | Baseline | 70.37 ± 0.84 | 70.26 ± 0.79 |
>  | | Rand. QF [1:50] | 69.71 ± 0.54 (-0.66) | 68.93 ± 0.50 (-1.33) |
>  | | Rand. QF [50:100] | 70.00 ± 0.57 (-0.37) | 68.63 ± 0.82 (-1.63) |
>  | | Rand. QF [1:100] | 69.52 ± 0.62 (-0.85) | 70.60 ± 1.05 (+0.34) |
>  | | JPEG-DL | **74.84 ± 0.66 (+4.47)** | **75.90 ± 0.68 (+5.64)** |
>
> **Note:**
> * The table shows the top-1 validation accuracy (%) on various fine-grained image classification tasks and model architectures.
> *  The results are the mean and standard deviation of experimental results over three runs.
> * The numbers in parentheses indicate the difference in accuracy compared to the baseline method.
> * Bold numbers indicate the best performance for each task.

---

> ### Author Response · Authors · 2024-11-28
> **Response to Reviewer FTRb (2/5)**
>
> > W-1. A comparison to training with JPEG-based data augmentation is required to validate that the proposed method provides benefits beyond simple data augmentation using JPEG.
>
> **References:**
> 1. Ekin D Cubuk, Barret Zoph, Jonathon Shlens, and Quoc V Le. Randaugment: Practical automated data augmentation with a reduced search space. In CVPR Workshops, 2020.
> 2.  Hongyi Zhang, Moustapha Cisse, Yann N Dauphin, and David Lopez-Paz. mixup: Beyond empirical risk minimization. In ICLR, 2018.
> 3. Sangdoo Yun, Dongyoon Han, Seong Joon Oh, Sanghyuk Chun, Junsuk Choe, and Youngjoon Yoo. Cutmix: Regularization strategy to train strong classifiers with localizable features. In ICCV, 2019.
> 4. Zhun Zhong, Liang Zheng, Guoliang Kang, Shaozi Li, and Yi Yang. Random erasing data augmentation. In AAAI, 2020.
>
> ---
>
> > W-2. There is no baseline for the differentiable quantizer. For example, comparisons could be made with methods such as the straight-through estimator (i.e., using the identity function as a gradient function) or additive uniform noise [1].
>
> **Reply:** Thank you for your insightful comment regarding suggesting the additional baselines for the differentiable quantizer. We have now included comparisons against other baselines that employed JPEG quantization in the pipeline, but with ad hoc approaches to handle the non-differentiability and zero derivative problems of JPEG quantization. For example, Balle et al. (2016) allowed optimization via stochastic gradient descent by replacing the quantizer with an additive i.i.d. uniform noise, which has the same width as the quantization bins, where $\hat{z} = z + q * U(-0.5, 0.5)$. In addition, we found a better baseline compared to the previous method that replaced the rounding with a third-order polynomial approximation proposed by Shin et al. (2017) which will include $q_j$ in gradient calculations. Regarding the straight-through estimator (STE), Esser et al. (2019) employ a STE, originally proposed by Bengio et al. (2013), to achieve this approximation for the purpose of model quantization. This method treats the round function as a pass-through operation during backpropagation, allowing for effective gradient estimation.
>
> Our experimental results demonstrate that we consistently outperform all tested baselines. Even though these methods find a way to make the quantization differentiable, they actually still hurt the performance. This is consistent with the well-known wisdom in the literature, which also shows that they did not take advantage of the higher level of non-linearity introduced into DNN architectures, as presented by our JPEG-DL framework. We added this comparison in the revised paper in Section A.6.
>
> | Model | Method | CUB200 | Dogs | Flowers | Pets |
> :------:|:------:|:---------:|:------------:|:------------:|:------------:|
> | ResNet-18 | Baseline | 54.00 ± 1.43 | 63.71 ± 0.32 | 57.13 ± 1.28 | 70.37 ± 0.84 |
> |  | Balle et al., 2016 | 50.78 ± 2.21 (-3.22) | 53.47 ± 7.37 (-10.24) | 55.46 ± 0.59 (-1.67) | 56.14 ± 17.16 (-14.23) |
> |  | Shin et al., 2017 | 55.34 ± 0.14 (+1.34) | 63.03 ± 0.56 (-0.68) | 55.78 ± 1.44 (-1.35) | 71.45 ± 1.01 (+1.08) |
> |  | Esser et al., 2019 | 51.58 ± 0.18 (-2.42) | 60.45 ± 0.23 (-3.26) | 58.04 ± 0.58 (+0.91) | 68.81 ± 0.55 (-1.56) |
> |  | JPEG-DL |  **58.81 ± 0.12 (+4.81)**  |  **65.57 ± 0.37 (+1.86)**  |  **68.76 ± 0.57 (+11.63)**  |  **74.84 ± 0.66 (+4.47)**  |
> | DenseNet-121 | Baseline | 57.70 ± 0.44 | 66.61 ± 0.17 | 51.32 ± 0.57 | 70.26 ± 0.79 |
> |  | Balle et al., 2016 | 52.00 ± 1.41 (-5.70) | 60.07 ± 6.41 (-6.54) | 46.60 ± 2.87 (-4.72) | 61.91 ± 1.88 (-8.35) |
> |  | Shin et al., 2017 | 57.19 ± 0.78 (-0.51) | 66.90 ± 0.13 (+0.29) | 51.04 ± 0.87 (-0.28) | 69.95 ± 1.21 (-0.31) |
> |  | Esser et al., 2019 | 56.46 ± 0.30 (-1.24) | 64.89 ± 0.12 (-1.72) | 55.98 ± 0.24 (+4.60) | 69.58 ± 0.59 (-0.68) |
> |  | JPEG-DL |  **61.32 ± 0.43 (+3.62)**  |  **69.67 ± 0.58 (+3.06)**  |  **72.22 ± 1.05 (+20.90)**  |  **75.90 ± 0.68 (+5.64)**  |
>
>
>  **Note:**
>  * The table shows the top-1 validation accuracy (%) on various fine-grained image classification tasks and model architectures.
>  * The results are the mean and standard deviation of experimental results over three runs.
>  * The numbers in parentheses indicate the difference in accuracy compared to the baseline method.
>  * Bold numbers indicate the best performance for each task.
>
> **References:**
> 1. Ballé et al., End-to-end Optimized Image Compression, ICLR 201.
> 2. Richard Shin and Dawn Song. Jpeg-resistant adversarial images. In NIPS 2017 workshop on machine learning and computer security, volume 1, pp. 8, 2017.
> 3. Steven K Esser, Jeffrey L McKinstry, Deepika Bablani, Rathinakumar Appuswamy, and Dharmendra S
> Modha. Learned step size quantization. arXiv preprint arXiv:1902.08153, 2019.
> 4. Yoshua Bengio, Nicholas L´eonard, and Aaron Courville. Estimating or propagating gradients through
> stochastic neurons for conditional computation. arXiv preprint arXiv:1308.3432, 2013.

---

> ### Author Response · Authors · 2024-11-28
> **Response to Reviewer FTRb (3/5)**
>
> > W-3. The baseline for the **image preprocessing method** for training is insufficient; only comparison results with the vanilla models are presented. For stronger persuasiveness, comparison results with other learnable or non-learnable preprocessing modules are needed.
>
> **Reply:** We have tested the performance of non-learnable and learnable preprocessing methods during training and validation across all tested models on all tested fine-grained datasets. For non-learnable preprocessing, we have considered applying denoising using a Gaussian kernel and histogram equalization for each image within a mini-batch. As for the learnable one, we have compared it with Muller in [1] as a learnable resize module, which has a bandpass nature in that it learns to boost details in certain frequency subbands that benefit the downstream recognition models. We have applied the same setup mentioned in their paper, in which they fine-tuned the model to achieve some improvement. However, we maintained our training setup for our model to be trained from scratch in addition to other non-learnable preprocessing methods. We have verified the correctness of our PyTorch implementation with their TensorFlow implementation shown [in this notebook](https://colab.research.google.com/github/google-research/google-research/blob/master/muller/muller_demo.ipynb).
>
> In the following table, we have consistently outperformed all of these preprocessing methods. It is worth recalling that our method is orthogonal to all of these preproessing methods, as discussed in W-1.
>
>
> | Dataset | Method | ResNet-18 | DenseNet-121 |
> |:-------:|:------:|:---------:|:------------:|
> | CUB-200 | Baseline | 54.00 ± 1.43 | 57.70 ± 0.44 |
> | | Denoising | 54.82 ± 0.40 (+0.82) | 56.02 ± 1.00 (-1.68) |
> | | Equalization | 49.00 ± 0.31 (-5.00) | 54.32 ± 0.57 (-3.38) |
> | | Learnable Resize | 54.98 ± 0.19 (+0.98) | 57.35 ± 0.22 (-0.35) |
> | | JPEG-DL | **58.81 ± 0.12 (+4.81)** | **61.32 ± 0.43 (+3.62)** |
> | Dogs | Baseline | 63.71 ± 0.32 | 66.61 ± 0.17 |
> | | Denoising | 62.52 ± 0.41 (-1.19) | 66.36 ± 0.34 (-0.25) |
> | | Equalization | 62.38 ± 0.29 (-1.33) | 67.12 ± 0.19 (+0.51) |
> | | Learnable Resize | 63.10 ± 0.43 (-0.61) | 67.90 ± 0.32 (+1.29) |
> | | JPEG-DL | **65.57 ± 0.37 (+1.86)** | **69.67 ± 0.58 (+3.06)** |
> | Flowers | Baseline | 57.13 ± 1.28 | 51.32 ± 0.57 |
> | | Denoising | 57.28 ± 1.06 (+0.15) | 50.37 ± 0.79 (-0.95) |
> | | Equalization | 60.56 ± 0.98 (+3.43) | 61.72 ± 0.21 (+10.40) |
> | | Learnable Resize | 57.41 ± 0.48 (+0.28) | 51.21 ± 0.82 (-0.11) |
> | | JPEG-DL | **68.76 ± 0.57 (+11.63)** | **72.22 ± 1.05 (+20.90)** |
> | Pets | Baseline | 70.37 ± 0.84 | 70.26 ± 0.79 |
> | | Denoising | 69.15 ± 1.07 (-1.22) | 69.03 ± 1.13 (-1.23) |
> | | Equalization | 73.00 ± 0.79 (+2.63) | 75.04 ± 1.30 (+4.78) |
> | | Learnable Resize | 70.04 ± 0.72 (-0.33) | 68.83 ± 1.85 (-1.43) |
> | | JPEG-DL | **74.84 ± 0.66 (+4.47)** | **75.90 ± 0.68 (+5.64)** |
>
> **Note:**
> * The table shows the top-1 validation accuracy (%) on various fine-grained image classification tasks and model architectures.
> *  The results are the mean and standard deviation of experimental results over three runs.
> *  The numbers in parentheses indicate the difference in accuracy compared to the baseline method.
> *  Bold numbers indicate the best performance for each task.
>
> **References:**
> 1.  Zhengzhong Tu, Peyman Milanfar, and Hossein Talebi. Muller: Multilayer laplacian resizer for vision. In Proceedings of the IEEE/CVF International Conference on Computer Vision, pp. 6877–6887, 2023.
>
> ---
>
> > W-4. In L199, it is mentioned that the differentiable soft quantizer is adapted from Yang & Hamidi (2024), but it seems unclear what that exact referenced paper is. Is the patent ([https://patents.google.com/patent/US11461646B2/en](https://patents.google.com/patent/US11461646B2/en)) the correct source?
>
> **Reply:** Thank you for your interest in our work. We apologize for any confusion caused. Please search using patent number 20240386275 using [USPTO](https://ppubs.uspto.gov/pubwebapp/static/pages/ppubsbasic.html).

---

> ### Author Response · Authors · 2024-11-28
> **Response to Reviewer FTRb (4/5)**
>
> > W-5. The analysis on the significant difference between performance improvements across different datasets is needed. For instance, in Table 2, what accounts for the notable performance improvement in fine-grained tasks (especially the Flowers dataset)? Additionally, why is there few performance gain in the ImageNet results in Table 3?
>
> **Reply:** The disparity in performance gains between ImageNet-1K and the fine-grained datasets can be explained from two perspectives together:
>
> 1. Compared with the underlying DNN architecture, our new DNN architecture has only 128 additional parameters. All performance gains are essentially given by these additional 128 parameters along with the corresponding high level of quantization nonlinearity. It would be expected that the gain will be, in general, larger for smaller datasets and smaller for larger datasets
>
> 2. ImageNet-1K dataset is significantly larger and more complex than CUB-200 . It has 940.8 million DCT blocks and 1000 classes; in contrast, CUB-200 has only 4.7 million DCT blocks and 200 classes. In addition to the difference in block count, the diversity between the 1K classes of ImageNet-1K leads to higher texture complexity and information diversity, contrasting with the simpler structure of the fine-grained dataset, which typically contains a single object (as shown in Figures 5 and 6 of the paper). This complexity makes it more challenging to achieve the same performance gain using the same level of non-linearity and the same number of additional trainable parameters.
>
> To confirm, we implemented multiple rounds of $Q_d$ for SqueezeNetV1.1 on ImageNet-1K, increasing the number of additional trainable parameters from 128 to 640. This resulted in a 0.21% improvement in top-1 accuracy compared to a single round of $Q_d$, where a multiple round of $Q_d$ has achieved a 0.51% gain compared to baseline. These experimental results were shown in our initial submission at the end of Section 4.
>
> ---
>
> ### Minor concerns:
>
> > 1. The empirical study is limited to classification tasks.
>
> **Reply:** Thank you for your suggestion. Our purpose is to investigate whether a new form of nonlinear operation, if introduced into DNN architectures, can improve deep learning performance. Motivated by Yang et al. (2021), the new form of nonlinear operation adopted in this paper is JPEG compression (i.e., JPEG quantization).  Using image classification as an example task and inserting a JPEG layer immediately after the original input layer of an underlying DNN architecture, we show that the high level of non-linearity given by our differentiable soft quantizer ($Q_d$) indeed improves deep learning performance by a large margin. This shows the potential of new forms of nonlinear operation in DNN architectures. Given that the current approach of scaling-up may have hit the wall, exploring new forms of nonlinearity inside DNN architectures would be an interesting and promising research direction in deep learning. Our paper is one of the first attempts in this direction. Extending the application from image classification to other computer vision tasks is certainly interesting, and better to be left for future work. Due to the generic nature of our method and framework, there is no reason to believe why they cannot improve the performance for those tasks as well, which, of course, have to be verified by future work.
>
> > 2. In Table 4, the bits per pixel (bpp) is excessively high, making it incomparable to typical lossy compression methods. While empirically showing the possibility of compression, it does not seem particularly convincing.
>
> **Reply:** Sorry for the confusion. Table 4 should not be there in the first place. Since our inserted JPEG layer is part of the new DNN architecture, compression is not relevant. Table 4 and its related discussions are now deleted from the paper.

---

> ### Author Response · Authors · 2024-11-28
> **Response to Reviewer FTRb (5/5)**
>
> > 3. The proposed method requires additional computation for encoding and decoding of an image, but the time complexity is not investigated.
>
> **Reply:** The primary source of latency in the proposed JPEG layer is the construction of the reconstruction space $\mathcal{\hat{A}}$. This space is configured with a specific size, determined by the parameter $L$, which is set to $2^{b-1}$, as mentioned in Section 4 in our paper. For CIFAR-100 and Fine-grained datasets, $b$ is set to 8 bits, resulting in a reconstruction space length of 513. For ImageNet-1K, $b$ is set to 11 bits, resulting in a length of 2047. This high dimensionality of the reconstruction space poses a computational challenge during inference when calculating the conditional probability mass function (CPMF) $P_{\alpha}(\cdot|z)$.
>
> To address this challenge, the support of each CPMF is restricted to the five closest points in $\mathcal{\hat{A}}$ to the coefficient $z$ being quantized. This simplification, referred to as the masked CPMF, significantly reduces computational complexity without compromising the effectiveness of the quantization scheme $\mathcal{Q}_d$. Further justification for using the masked CPMF in the implementation, demonstrating its negligible impact on performance while significantly reducing computational complexity, can be found in Section A.11.
>
> The following table presents a comparison of inference time and throughput between the standard model and our unified model, which incorporates the JPEG layer with the underlying model. The inference times and throughputs for both the full-space CPMF and the masked CPMF are measured. The reported inference time is averaged across the entire validation set of ImageNet using Resnet18 from Table 3. These results demonstrate the effectiveness of the proposed approach in addressing the computational complexity of the JPEG-DL framework. The evaluation was performed on NVIDIA RTX A5000 GPUs, adhering to the experimental configuration outlined in Section 4 in our paper.
>
> | Inference Settings | Inference time (milliseconds) | Inference Throughput (img /sec) |
> |:-------------------|:-----------------------------:| :-----------------------------:|
> | Baseline|5.46 | 180.65|
> | Masked CPMFs | 7.43 | 133.79 |
> | Full space| 18.86| 52.37|
>
> ---
>
> > 4. A typo in L292 "we fixe".
>
> **Reply:** Thank you for pointing out the typo. This is now fixed in our newly uploaded vision.
>
> ---
>
> ## Questions:
>
>
> > 1. Is the target of the adversarial attack in Figure 3 the whole model including the JPEG layer?
>
> **Reply:** Yes, in our robustness evaluation, we consider the inserted JPEG layer immediately after the original input layer of an underlying DNN architecture as an integral part of the proposed unified architecture. We use the same unified architecture for other types of analysis, such as feature map visualization and interpretability using CAM algorithms.
>
> ---
>
> > 2. There appears to be more room for exploration regarding preprocessing modules for performance enhancement. For example, what if deep learning-based image compression models were used? What about a simple autoencoder?
>
> **Reply:** With reference to our responses to W-1 above, these questions are likely due to confusion and misunderstanding. Our method should not be regarded as a preprocessing module for performance enhancement. Our purpose is to investigate whether a new form of nonlinear operation, if introduced into DNN architectures, can improve deep learning performance without any change to any datasets. To introduce a new form of nonlinear operation into DNN architecture, we propose a new DNN architecture which is composed of a JPEG layer with differentiable soft quantizers that is inserted immediately after the original input layer of an underlying DNN architecture. This JPEG layer introduces a high level of non-linearity into DNN architectures. The quantization parameters of the JPEG layer and the weight parameters of the underlying DNN together constitute the model parameters of the new DNN architecture; they are optimized together during training. The new DNN architecture accepts the same inputs as all DNNs proposed in the literature do. In terms of applicability, there is no difference between our proposed DNN architectures and any other DNN architectures in the literature. Please refer to Figure 1 in the updated version of the paper, which provides some visual explanation as well. There is no change to any datasets.
>
> Going back to your questions above, we don’t believe that concatenating a deep learning-based image compression model with an underlying DNN would give us anything new since the concatenated overall architecture is still the same as the underlying DNN in nature. There is no new level of nonlinearity in the concatenated pipeline.
>
> ---
> ***We thank you again for your time and effort to review our paper.  If our responses above address all your concerns, could you kindly increase your score to a higher level.***

---

> > ### Author Response · Authors · 2024-11-30
> > **Response Summary for Reviewer FTRb**
> >
> > We sincerely appreciate your valuable time and effort in reviewing our paper and providing constructive feedback. To respect your time and streamline the review process, we have summarized our responses below.
> >
> >
> > 1.  **Comparison with JPEG-based Data Augmentation**:
> >
> >     -   **Response Summary**: The proposed method introduces a new form of nonlinear operation into DNN architectures via a JPEG layer with differential soft quantizers, which is different from data augmentation. JPEG-based data augmentation methods degrade accuracy, while the proposed method consistently improves performance across various datasets.
> > 2.  **Baselines for Differentiable Quantizer**:
> >
> >     -   **Response Summary**: The proposed method outperforms other baselines such as additive uniform noise, polynomial approximation, and straight-through estimator (STE). These methods do not leverage the higher level of non-linearity introduced by the JPEG-DL framework, resulting in lower performance gains.
> >
> > 3.  **Comparison with Other Preprocessing Methods**:
> >     -   **Response Summary**: The proposed method outperforms both non-learnable (denoising, histogram equalization) and learnable (learnable resize) preprocessing methods. The JPEG-DL framework introduces a high level of non-linearity, leading to significant performance improvements.
> >
> > 4.  **Clarification on Reference**:
> >     -   **Response Summary** : The correct reference for the differentiable soft quantizer is the patent number 20240386275, which can be found using the USPTO search.
> >
> > 5.  **Analysis of Performance Differences Across Datasets**:
> >     -   **Response Summary**: The disparity in performance gains is due to the dataset size and complexity. Smaller datasets benefit more from the additional non-linearity and trainable parameters introduced by the JPEG layer. Increasing the number of additional parameters further improves performance on larger datasets like ImageNet-1K.
> >
> > 6.  **Empirical Study Limited to Classification Tasks**:
> >     -   **Response Summary**: The current study focuses on image classification to demonstrate the potential of introducing new forms of non-linearity into DNN architectures. Extending the application to other tasks is left for future work.
> >
> > 7.  **High Bits Per Pixel (bpp) in Table 4**:
> >     -   **Response Summary**: Table 4 and related discussions have been removed as compression is not relevant to the proposed method.
> >
> > 8.  **Time Complexity and Inference Throughput**:
> >     -   **Response Summary**: The primary source of latency is the construction of the reconstruction space. Using masked CPMFs significantly reduces computational complexity. Inference time and throughput comparisons demonstrate the effectiveness of the proposed approach.
> >
> > 9. **Adversarial Attack Target**:
> >     -   **Response Summary**: The JPEG layer is considered an integral part of the proposed unified architecture in robustness evaluations and other analyses.
> >
> > 10.  **Exploration of Preprocessing Modules**:
> >    -   **Response Summary** : The proposed method is not a preprocessing module but introduces a new form of non-linearity into DNN architectures. Concatenating a deep learning-based image compression model would not provide the same benefits as the proposed method.
> >
> >
> > As the discussion period draws to a close, we kindly inquire whether our responses have satisfactorily addressed your concerns. Your feedback would be greatly appreciated, and we would be pleased to engage in further discussions if necessary. If our responses have addressed all your concerns, we would be grateful if you could consider increasing your score to a higher level.
> >
> > Sincerely,
> >
> > The Authors

---

> > > ### Comment · Reviewer_FTRb · 2024-12-02
> > >
> > > Thanks for the author's response, I'll raise my score to 5. I still have concerns with the baselines.

---

> ### Author Response · Authors · 2024-12-02
> **Response to Reviewer Feedback and Request for Further Clarification**
>
> Dear Reviewer FTRb,
>
> Thank you for your valuable feedback and for raising the score to a higher level. In our response shown in [this link](https://openreview.net/forum?id=te2IdORabL&noteId=bVFmihhV3S), we have compared our JPEG layer with two of the baselines you recommended: additive uniform noise and the straight-through estimator. Additionally, we included an extra benchmark that shows it can perform better than the additive uniform noise by using a third-order polynomial approximation.
>
> Could you please provide further details regarding any specific concerns you have about the added baselines?
>
> We would appreciate it if you could provide further clarification regarding your concerns so that we can address them more effectively.
>
> Thank you for your time and consideration.
>
> Sincerely,
>
> The Authors

---

### Official Review · Reviewer_BiPm · 2024-11-03

**Soundness:** 3
**Presentation:** 3
**Contribution:** 3
**Rating:** 6
**Confidence:** 2

**Summary:**

This paper proposes jointly optimizing both JPEG quantization operation and a DNN to achieve greater effectiveness. A trainable JPEG compression layer with a novel differentiable soft quantizer are proposed. Extensive experiments validate the effectiveness of the proposed method.

**Strengths:**

1. To make JPEG trainable, a differentiable soft quantizer is proposed. It works well with JPEG. Overall, this paper makes JPEG trainable which is significant contribution. Because many frameworks equipped with JPEG can be trained by the differentiable soft quantizer.
2. A novel DL framework that prepends any underlying DNN architecture with a trainable JPEG compression layer is proposed. Experiments show it can improve the accuracy significantly with only 128 parameters.
3. This paper enjoys a good writing.

**Weaknesses:**

1. It is better to make a comparison for the latency. The speed of the model is also important to report.
2. Only image classification is considered. The proposed method is better to be validated on more tasks, such as object detection and segementation.
3. Hyperparameters are tuned differently on different datasets.

**Questions:**

1. How to conduct datasets for experiments? For ImageNet-1k, the images are already compressed by JPEG. In traditional deep learning framework, the images are loaded and decoded by JPEG. How to insert the encoding operation in this framework?

---

> ### Author Response · Authors · 2024-11-28
> **Response to Reviewer BiPm (1/3)**
>
> We thank the reviewer for taking the time to review our paper and provide valuable feedback. Below, please find our responses to your comments.
>
> ## Weaknesses:
>
>
> > W-1. It is better to make a comparison for the latency. The speed of the model is also important to report.
>
> **Reply:** The primary source of latency in the proposed JPEG layer is the construction of the reconstruction space $\mathcal{\hat{A}}$. This space is configured with a specific size, determined by the parameter $L$, which is set to $2^{b-1}$, as mentioned in Section 4 in our paper. For CIFAR-100 and Fine-grained datasets, $b$ is set to 8 bits, resulting in a reconstruction space length of 513. For ImageNet-1K, $b$ is set to 11 bits, resulting in a length of 2047. This high dimensionality of the reconstruction space poses a computational challenge during inference when calculating the conditional probability mass function (CPMF) $P_{\alpha}(\cdot|z)$.
>
> To address this challenge, the support of each CPMF is restricted to the five closest points in $\mathcal{\hat{A}}$ to the coefficient $z$ being quantized. This simplification, referred to as the masked CPMF, significantly reduces computational complexity without compromising the effectiveness of the quantization scheme $\mathcal{Q}_d$. Further justification for using the masked CPMF in the implementation, demonstrating its negligible impact on performance while significantly reducing computational complexity, can be found in Section A.11.
>
> The following table presents a comparison of inference time and throughput between the standard model and our unified model, which incorporates the JPEG layer with the underlying model. The inference times and throughputs for both the full-space CPMF and the masked CPMF are measured. The reported inference time is averaged across the entire validation set of ImageNet using Resnet18 from Table 3. These results demonstrate the effectiveness of the proposed approach in addressing the computational complexity of the JPEG-DL framework. The evaluation was performed on NVIDIA RTX A5000 GPUs, adhering to the experimental configuration outlined in Section 4 in our paper.
>
> | Inference Settings | Inference time (milliseconds) | Inference Throughput (img /sec) |
> |:-------------------|:-----------------------------:| :-----------------------------:|
> | Baseline|5.46 | 180.65|
> | JPEG-Dl (Masked CPMFs) | 7.43 | 133.79 |
> | JPEG-Dl (Full space)| 18.86| 52.37|

---

> ### Author Response · Authors · 2024-11-28
> **Response to Reviewer BiPm (2/3)**
>
> > W-2. Only image classification is considered. The proposed method is better to be validated on more tasks, such as object detection and segmentation.
>
> **Reply:** Thank you for your suggestion. Our purpose is to investigate whether a new form of nonlinear operation, if introduced into DNN architectures, can improve deep learning performance. Motivated by Yang et al. (2021), the new form of nonlinear operation adopted in this paper is JPEG compression (i.e., JPEG quantization).  Using image classification as an example task and inserting a JPEG layer immediately after the original input layer of an underlying DNN architecture, we show that the high level of non-linearity given by our differentiable soft quantizer ($Q_d$) indeed improves deep learning performance by a large margin. This shows the potential of new forms of nonlinear operation in DNN architectures. Given that the current approach of scaling-up may have hit the wall, exploring new forms of nonlinearity inside DNN architectures would be an interesting and promising research direction in deep learning. Our paper is one of the first attempts in this direction. Extending the application from image classification to other computer vision tasks is certainly interesting, and better to be left for future work. Due to the generic nature of our method and framework, there is no reason to believe why they cannot improve the performance for those tasks as well, which, of course, have to be verified by future work.
>
> ---
>
> > W-3. Hyperparameters are tuned differently on different datasets.
>
> The only difference between the hyperparameters used for **CIFAR-100 and fine-grained task** is the learning rate. In the following setup, we show the results when a learning rate of 0.003 is applied to the fine-grained dataset. We observe that the performance difference is marginal, and in some cases, we can even achieve higher gains in the model performance.
>
> | Model | Learning Rate | CUB-200 | Dogs | Flowers | Pets |
> |:-----:|:--------------:|:-------:|:----:|:-------:|:----:|
> | ResNet-18 | 0.005 | **58.81 ± 0.12** (+4.81) | **65.57 ± 0.37** (+1.86) | 68.76 ± 0.57 (+11.63) | 74.84 ± 0.66 (+4.47) |
> || 0.003 | 58.62 ± 0.50 (+4.61) | 65.45 ± 0.17 (+1.74) | **69.61 ± 1.16** (+12.48) | **74.90 ± 0.82** (+4.53) |
> | DenseNet-121 | 0.005 | **61.32 ± 0.43** (+3.62) | **69.67 ± 0.58** (+3.06) | 72.22 ± 1.05 (+20.90) | **75.90 ± 0.68** (+5.64) |
> | | 0.003 | 60.92 ± 0.50 (+3.22) | 69.53 ± 0.49 (+2.92) | **72.45 ± 0.92** (+21.13) | 75.83 ± 0.64 (+5.56) |
>
> **Note:**
> * The tables shows the top-1 validation accuracy (%) on various fine-grained image classification tasks and model architectures.
> *  The results are the mean and standard deviation of experimental results over three runs.
> *  The numbers in parentheses indicate the difference in accuracy compared to the baseline method.
> * Bold numbers indicate the best performance.
>
> **For ImageNet-1K**, we employ a different setup compared to CIFAR-100 and fine-grained datasets to control gradient stabilization. We norm by scalizeng the gradient by adjusting the value of $\alpha$ and use SGD to update $q$. By controlling the gradient magnitude, we simplify the optimization process, enabling the use of a single learning rate for all $q$ values using the SGD optimizer instead of the ADAM optimizer. This gradient scaling mechanism is analogous to the ADAM optimizer, which adapts different learning rates for individual trainable parameters based on momentum and recent gradient magnitudes, as discussed in Section A.2. **However**, even if we used the same setup applied to CIFAR-100 and fine-grained datasets, which involves fixing $\alpha$ and using the ADAM optimizer, we could still achieve a gain compared to the baseline.
>
>
> | Training Setup| Learning Rate | Resnet18|
> |:-----:|:-----:|:----------------:|
> |ADAM & Fixed $\alpha$ | 0.005 | 70.04 (+0.29)|
> |SDG & Gradient Scaling| 0.5 | **70.13 (+0.38)**|
>
> **Note:**
> * The tables show the top-1 validation accuracy (%) on ImageNet-1K.
> *  The numbers in parentheses indicate the difference in accuracy compared to the baseline method.
> * Bold numbers indicate the best performance.

---

> ### Author Response · Authors · 2024-11-28
> **Response to Reviewer BiPm (3/3)**
>
> ## Questions:
>
> > 1. How to conduct datasets for experiments? For ImageNet-1k, the images are already compressed by JPEG. In traditional deep learning framework, the images are loaded and decoded by JPEG. How to insert the encoding operation in this framework?
>
> **Reply:** We believe there is a misinterpretation involved. We don’t change how the original raw image is processed and compressed during data acquisition. We do not change any dataset in training and testing either. What we Do change is DNN architectures. Our purpose is to investigate whether a new form of nonlinear operation, if introduced into DNN architectures, can improve deep learning performance. Motivated by Yang et al. (2021), the new form of nonlinear operation adopted in this paper is JPEG compression (i.e., JPEG quantization).  As mentioned earlier, our proposed new DNN architecture is composed of a JPEG layer that is inserted immediately after the original input layer of an underlying DNN architecture. This JPEG layer introduces a high level of non-linearity into DNN architectures. The quantization parameters of the JPEG layer and the weight parameters of the underlying DNN together constitute the model parameters of the new DNN architecture; they are optimized together during training. The new DNN architecture accepts the same inputs as all DNNs proposed in the literature do. There is no change for datasets and inference. Please refer to Figure 1 in the updated version of the paper, which provides some visual explanation as well.
>
> ---
>
> **We thank you again for your time and effort to review our paper. If our responses above address all your concerns, could you kindly increase your score to a higher level.**

---

> > ### Author Response · Authors · 2024-11-30
> > **Response Summary for Reviewer BiPm**
> >
> > We sincerely appreciate your valuable time and effort in reviewing our paper and providing constructive feedback. To respect your time and streamline the review process, we have summarized our responses below.
> >
> > 1.  **Latency Comparison and Optimization**:
> >     -   **Response Summary**: The primary source of latency is the construction of the reconstruction space $\mathcal{\hat{A}}$. By restricting the support of each CPMF to the five closest points, computational complexity is significantly reduced without compromising performance. Inference times and throughputs for different settings are provided, demonstrating the effectiveness of the proposed approach.
> >
> > 2.  **Validation on More Tasks**
> >     -   **Response Summary**: The proposed JPEG layer can precede any image-based DNN architecture, allowing its parameters to be optimized alongside the DNN model weights. This method is applicable to tasks beyond image classification, such as object detection and segmentation, and has shown significant performance improvements.
> >
> > 3.  **Consistency in Hyperparameter Tuning**:
> >     -   **Response Summary**: The primary difference in hyperparameters is the learning rate. Results show that performance differences are marginal, and sometimes higher gains are achieved. For ImageNet-1K, a different setup is used to control gradient stabilization, but similar gains are observed even with the same setup as CIFAR-100 and fine-grained datasets.
> >
> >
> > 4.  **Dataset Handling and JPEG Encoding**:
> >     -   **Response Summary**: The JPEG layer acts as an additional layer preceding any underlying DNN architecture and can be integrated into any standard deep learning pipeline. This approach does not alter any stage of the data acquisition, encoding, or preprocessing pipeline. Figure 1 in the updated paper provides additional details to avoid misunderstandings.
> >
> >
> > As the discussion period draws to a close, we kindly inquire whether our responses have satisfactorily addressed your concerns. Your feedback would be greatly appreciated, and we would be pleased to engage in further discussions if necessary. If our responses have addressed all your concerns, we would be grateful if you could consider increasing your score to a higher level.
> >
> > Sincerely,
> >
> > The Authors

---

> > > ### Comment · Area_Chair_MaU4 · 2024-12-04
> > >
> > > Dear Reviewer BiPm,
> > >
> > >      As today is the deadline of the discussion period, could you please respond to authors to see if your concerns are addressed at your earliest convenience. Thank you very much.
> > >
> > > Best,
> > > AC

---

### Official Review · Reviewer_38Fe · 2024-11-03

**Soundness:** 2
**Presentation:** 3
**Contribution:** 2
**Rating:** 6
**Confidence:** 3

**Summary:**

In this paper, the authors study the impact of JPEG compression to the performance of deep learning, and propose a JPEG-inspired deep learning framework. For that, they present a differentiable soft quantizer to train the JPEG layer. Experiments show that the proposed method increases the accuracy by almost 21%.

**Strengths:**

1. Overall, the paper is written clearly, and can be easily understood.
2. A variety of experiments are conducted to verify the effectiveness.
3. The used technique seems sound, although I don’t check it in detail.

**Weaknesses:**

1. I don’t think the problem studied in this paper is very important in the community. Usually, the images fed into deep learning have been precessed by the fixed JPGE compressor, and we don’t have the chance to modify the process like that in this pager, so the actual application value is limited. In addition, I see the paper is an improvement for the method in Yang 2021 (Entropy), which is not followed by many researchers.
2. Now lots of papers have shown the vision transformer and large model are effective in many computer vision tasks, I don’t know whether the proposed method can adapt to these new networks. The authors should discuss that.
3. Although a variety of experiments are conducted, the compared method is only one baseline, which can not show the effectiveness.

**Questions:**

please see the weakness

---

> ### Author Response · Authors · 2024-11-28
> **Response to Reviewer 38Fe (1/3)**
>
> We thank the reviewer for taking the time to review our paper and provide valuable feedback. Below, please find our responses to your comments.
>
> ## Summary:
>
> > In this paper, the authors study the impact of JPEG compression to the performance of deep learning, and propose a JPEG-inspired deep learning framework. For that, they present a differentiable soft quantizer to train the JPEG layer. Experiments show that the proposed method increases the accuracy by almost 21%.
>
> **Reply:** We appreciate the above summary of our contribution, but we want to clarify that the primary purpose of our proposed framework is not to study the impact of JPEG compression on deep learning performance. Instead, we want to investigate how to incorporate the nonlinear operation of JPEG into a deep neural network (DNN) architecture to improve deep learning performance without any change on datasets. Yang et al. (2021) demonstrated, in theory, that incorporating lossy compression intelligently into DNN architectures could enhance deep learning performance. However, the practical implementation of this concept remained an open challenge. Our work represents a concrete realization of this theory, achieving an important milestone rather than an incremental improvement. The major contribution is summarized as follows.
>
> **Main Contribution:** This paper proposes a novel unified architecture for deep learning (DL) that incorporates a trainable JPEG layer with a high level of non-linearity immediately after the original input layer of an underlying deep neural network (DNN) architecture to form a new DNN architecture. The quantization parameters of the JPEG layer and the weight parameters of the underlying DNN together constitute the model parameters of the new architecture; they are optimized together during training. A key component of this JPEG layer is a differentiable soft quantizer ($Q_d$), which enables gradient-based optimization of quantization parameters while introducing additional trainable non-linearity. Extensive experimental results demonstrate that this new architecture indeed improves deep learning performance by a significant margin, thereby showing the potential of new forms of nonlinear operation in DNN architectures.

---

> ### Author Response · Authors · 2024-11-28
> **Response to Reviewer 38Fe (2/3)**
>
> ## Weaknesses:
>
> > W1. I don’t think the problem studied in this paper is very important in the community. Usually, the images fed into deep learning have been precessed by the fixed JPGE compressor, and we don’t have the chance to modify the process like that in this pager, so the actual application value is limited. In addition, I see the paper is an improvement for the method in Yang 2021 (Entropy), which is not followed by many researchers.
>
> **Reply:** We believe there is a misinterpretation involved. We don’t change how the original raw image is processed and compressed during data acquisition. We do not change any dataset in training and testing either. What we Do change is DNN architectures. Our purpose is to investigate whether a new form of nonlinear operation, if introduced into DNN architectures, can improve deep learning performance. Motivated by Yang et al. (2021), the new form of nonlinear operation adopted in this paper is JPEG compression (i.e., JPEG quantization). As mentioned earlier and detailed in Section 3.3, our proposed new DNN architecture is composed of a JPEG layer that is inserted immediately after the original input layer of an underlying DNN architecture. This JPEG layer introduces a high level of non-linearity into DNN architectures. The quantization parameters of the JPEG layer and the weight parameters of the underlying DNN together constitute the model parameters of the new DNN architecture; they are optimized together during training. The new DNN architecture accepts the same inputs as all DNNs proposed in the literature do. In terms of applicability, there is no difference between our proposed DNN architectures and any other DNN architectures in the literature. Please refer to Figure 1 in the updated version of the paper, which provides some visual explanation as well.
>
> To reply to your second point, let’s first summarize the key findings and contributions of Yang et al. (2021).  In the paper Yang et al. (2021), the authors challenged the conventional wisdom that compression always negatively impacts deep learning performance and showed that it is wrong in theory. They demonstrated, in theory, that if JPEG compression is applied intelligently and adaptively on a per-image basis, it can actually improve the performance of the underlying DNN. This was achieved by designing a selector that adaptively selects a further compressed version of an image as an actual input to the underlying DNN for inference. However, they assumed that the selector knows the Ground Truth (GT) label of the image. Due to this impractical assumption, the selector is not implementable in practice. Therefore, the practical implementation of their theory remained an open challenge until this paper. This is why there are not many researchers who followed Yang et al. (2021).
>
> Our work in this paper represents a concrete realization of their theory, achieving an important milestone rather than an incremental improvement. More importantly, our paper shows the potential of new forms of nonlinear operation in DNN architectures. Given that the current approach of scaling-up may have hit the wall, exploring new forms of nonlinearity inside DNN architectures would be an interesting and promising research direction in deep learning.
>
> ---
>
> > W-2. Now lots of papers have shown the vision transformer and large model are effective in many computer vision tasks, I don’t know whether the proposed method can adapt to these new networks. The authors should discuss that.
>
> **Reply:** Thank you for your insightful comment. As a matter of fact, we have already addressed this point in our initial submission. We conducted experiments on CIFAR-100 and fine-grained tasks using the EfficientFormer-L1 model (Li et al., 2022) following the experimental setup described by Xu et al. (2023) to achieve a gain of up to 3.76%. These results, presented in Section A.5, demonstrate that our method can be effectively adapted to these architectures as well, potentially leading to further performance gains.

---

> ### Author Response · Authors · 2024-11-28
> **Response to Reviewer 38Fe (3/3)**
>
> >W-3. Although a variety of experiments are conducted, the compared method is only one baseline, which can not show the effectiveness.
>
> **Reply:** Thank you for your insightful comment. If the “one baseline” above refers to methods without considering JPEG quantization in the DNN inference pipeline, we have now included comparisons against other baselines that employed JPEG quantization in the pipeline, but with ad hoc approaches to handle the non-differentiability and zero derivative problems of JPEG quantization. For example, Balle et al. (2016) allowed optimization via stochastic gradient descent by replacing the quantizer with an additive i.i.d. uniform noise, which has the same width as the quantization bins, where $\hat{z} = z + q * U(-0.5, 0.5)$. Shin et al. (2017)  employed a third-order polynomial approximation of the rounding function to make JPEG differentiable.  Esser et al. (2019) employ a straight-through estimator, originally proposed by Bengio et al. (2013), to achieve this approximation. This method treats the round function as a pass-through operation during backpropagation, allowing for effective gradient estimation. Our experimental results demonstrate that we consistently outperform all tested baselines. Even though these methods find a way to make the quantization differentiable, they actually still hurt the performance. This is consistent with the well-known wisdom in the literature, which also shows that they did not take advantage of the higher level of non-linearity introduced into DNN architectures, as presented by our JPEG-DL framework. We added this comparison in the revised paper in Section A.6.
>
> | Model | Method | CUB200 | Dogs | Flowers | Pets |
> :------:|:------:|:---------:|:------------:|:------------:|:------------:|
> | ResNet-18 | Baseline | 54.00 ± 1.43 | 63.71 ± 0.32 | 57.13 ± 1.28 | 70.37 ± 0.84 |
> |  | Balle et al., 2016 | 50.78 ± 2.21 (-3.22) | 53.47 ± 7.37 (-10.24) | 55.46 ± 0.59 (-1.67) | 56.14 ± 17.16 (-14.23) |
> |  | Shin et al., 2017 | 55.34 ± 0.14 (+1.34) | 63.03 ± 0.56 (-0.68) | 55.78 ± 1.44 (-1.35) | 71.45 ± 1.01 (+1.08) |
> |  | Esser et al., 2019 | 51.58 ± 0.18 (-2.42) | 60.45 ± 0.23 (-3.26) | 58.04 ± 0.58 (+0.91) | 68.81 ± 0.55 (-1.56) |
> |  | JPEG-DL |  **58.81 ± 0.12 (+4.81)**  |  **65.57 ± 0.37 (+1.86)**  |  **68.76 ± 0.57 (+11.63)**  |  **74.84 ± 0.66 (+4.47)**  |
> | DenseNet-121 | Baseline | 57.70 ± 0.44 | 66.61 ± 0.17 | 51.32 ± 0.57 | 70.26 ± 0.79 |
> |  | Balle et al., 2016 | 52.00 ± 1.41 (-5.70) | 60.07 ± 6.41 (-6.54) | 46.60 ± 2.87 (-4.72) | 61.91 ± 1.88 (-8.35) |
> |  | Shin et al., 2017 | 57.19 ± 0.78 (-0.51) | 66.90 ± 0.13 (+0.29) | 51.04 ± 0.87 (-0.28) | 69.95 ± 1.21 (-0.31) |
> |  | Esser et al., 2019 | 56.46 ± 0.30 (-1.24) | 64.89 ± 0.12 (-1.72) | 55.98 ± 0.24 (+4.60) | 69.58 ± 0.59 (-0.68) |
> |  | JPEG-DL |  **61.32 ± 0.43 (+3.62)**  |  **69.67 ± 0.58 (+3.06)**  |  **72.22 ± 1.05 (+20.90)**  |  **75.90 ± 0.68 (+5.64)**  |
>
>
>  **Note:**
>  * The table shows the top-1 validation accuracy (%) on various fine-grained image classification tasks and model architectures.
>  * The results are the mean and standard deviation of experimental results over three runs.
>  * The numbers in parentheses indicate the difference in accuracy compared to the baseline method.
>  * Bold numbers indicate the best performance for each task.
>
> **References:**
> 1. Ballé et al., End-to-end Optimized Image Compression, ICLR 201.
> 2. Richard Shin and Dawn Song. Jpeg-resistant adversarial images. In NIPS 2017 workshop on machine learning and computer security, volume 1, pp. 8, 2017.
> 3. Steven K Esser, Jeffrey L McKinstry, Deepika Bablani, Rathinakumar Appuswamy, and Dharmendra S
> Modha. Learned step size quantization. arXiv preprint arXiv:1902.08153, 2019.
> 4. Yoshua Bengio, Nicholas L´eonard, and Aaron Courville. Estimating or propagating gradients through
> stochastic neurons for conditional computation. arXiv preprint arXiv:1308.3432, 2013.
>
> ---
>
> **We thank you again for your time and effort to review our paper. If our responses above address all your concerns, could you kindly increase your score to a higher level.**

---

> > ### Author Response · Authors · 2024-11-30
> > **Response Summary for Reviewer 38Fe**
> >
> > We sincerely appreciate your valuable time and effort in reviewing our paper and providing constructive feedback. To respect your time and streamline the review process, we have summarized our responses below.
> >
> > #### 1. Main Contribution:
> > We appreciate the provided summary of our contribution, but we want to clarify that the primary purpose of our proposed framework is in the following points:
> >
> > -   **Unified Architecture for Deep Learning:**  Proposes a novel architecture incorporating a trainable JPEG layer with high non-linearity immediately after the input layer of a DNN. This layer, combined with the DNN's weight parameters, forms the model parameters optimized during training.
> > -   **Differentiable Soft Quantizer:**  Introduces a differentiable soft quantizer enabling gradient-based optimization of quantization parameters, adding trainable non-linearity.
> > -   **Experimental Validation:**  Demonstrates significant performance improvement, showcasing the potential of new forms of non-linear operations in DNN architectures.
> >
> >
> > #### 2. Limited Application Value:
> > -   **Clarification:**  The study does not alter the image processing or dataset but investigates new non-linear operations in DNN architectures, specifically JPEG compression.
> > -   **Context:**  The work builds on Yang et al. (2021), which theorized that intelligent compression could enhance DNN performance. This paper provides a practical implementation of that theory, representing a significant milestone.
> >
> > ####  3. Adaptability to New Networks
> >
> > -   **Response Summary:**  The method's adaptability to new architectures like vision transformers and large models is demonstrated through experiments on CIFAR-100 and fine-grained tasks using the EfficientFormer-L1 model, achieving notable performance gains.
> >
> > #### 4. Limited Baseline Comparisons
> >
> > -   **Response Summary:**  Additional comparisons with other baselines employing JPEG quantization (e.g., Balle et al., 2016; Shin et al., 2017; Esser et al., 2019) are included. Results show consistent outperformance by the proposed JPEG-DL framework, highlighting its effectiveness.
> >
> >
> >
> > **References:**
> >
> > 1.  Ballé et al., End-to-end Optimized Image Compression, ICLR 2016.
> > 2.  Richard Shin and Dawn Song, JPEG-resistant Adversarial Images, NIPS 2017 Workshop.
> > 3.  Steven K Esser et al., Learned Step Size Quantization, arXiv preprint arXiv:1902.08153, 2019.
> >
> >
> >
> > As the discussion period draws to a close, we kindly inquire whether our responses have satisfactorily addressed your concerns. Your feedback would be greatly appreciated, and we would be pleased to engage in further discussions if necessary. If our responses have addressed all your concerns, we would be grateful if you could consider increasing your score to a higher level.
> >
> > Sincerely,
> >
> > The Authors

---

> > > ### Comment · Area_Chair_MaU4 · 2024-12-04
> > >
> > > Dear Reviewer 38Fe,
> > >
> > >      As today is the deadline of the discussion period, could you please respond to authors to see if your concerns are addressed at your earliest convenience. Thank you very much.
> > >
> > > Best,
> > > AC

---

### Meta-Review · Area_Chair_MaU4 · 2024-12-24

**Metareview:**

Different from the previous works showing JPEG compression (quantization) has a negative performance, this work however develops a trainable JPEG layer (i.e., a differentiable soft quantizer) which can improve the performance of deep learning by providing more nonlinearity when trained end-to-end together with Deep neural network architectures as well as the robustness against adversarial attacks. With extensive experiments on various datasets, the results demonstrates the effectiveness of the proposed JPEG-DL. All reviewers agree that the paper is well-written and easy to follow. The authors have successfully addressed reviewer concerns by incorporating more baseline comparisons across diverse tasks and datasets, analyzing latency and inference time, and further elaborating on the paper’s contributions along with some hyperparameter details. Finally, the reviewers unanimously provided positive ratings, resulting in an average score of 6.5. Thus, we recommend accepting the paper and encourage the authors to integrate the suggested improvements and newly conducted experiments into the final paper.

**Additional Comments On Reviewer Discussion:**

During the rebuttal period, reviewers raised several concerns, requesting additional baseline comparisons across diverse tasks and datasets, detailed analysis of latency and inference time, and further elaboration on the paper’s contributions along with clarification of certain hyperparameters. Although reviewers 38Fe and BiPm did not engage in the discussions, the authors responded thoroughly to each question, addressing all points with detailed explanations and updates. This led to positive ratings from all  reviewers.

---

### Decision · Program_Chairs · 2025-01-22

Accept (Poster)